# Genome-wide association meta-analysis of age at onset of walking in over 70,000 infants of European ancestry

Anna Gui [1,2], Anja Hollowell [2], Emilie M. Wigdor [3], Morgan J. Morgan [4], Laurie J. Hannigan [5,6,7], Elizabeth C. Corfield [5,6], Veronika Odintsova [8,9], Jouke-Jan Hottenga [8], Andrew Wong [10], René Pool [8], Harriet Cullen [11,12], Siân Wilson [13,14], Varun Warrier [15], Espen M. Eilertsen [16], Ole A. Andreassen [17,18], Christel M. Middeldorp [19,20,21,22,23], Beate St Pourcain [24,25,26], Meike Bartels [8], Dorret I. Boomsma [27], Catharina A. Hartman [28], Elise B. Robinson [29], Tomoki Arichi [11], Anthony D. Edwards [11], Mark H. Johnson [2,30], Frank Dudbridge [31], Stephan J. Sanders [3,32,33], Alexandra Havdahl [5,6,16,33] & Angelica Ronald [2,4,33] ✉

Age at onset of walking is an important early childhood milestone which is used clinically and in public health screening. In this genome-wide association study meta-analysis of age at onset of walking ($N$ = 70,560 European-ancestry infants), we identified 11 independent genome-wide significant loci. SNP-based heritability was 24.13% (95% confidence intervals = 21.86–26.40) with ~11,900 variants accounting for about 90% of it, suggesting high polygenicity. One of these loci, in gene *RBL2*, co-localized with an expression quantitative trait locus (eQTL) in the brain. Age at onset of walking (in months) was negatively genetically correlated with ADHD and body-mass index, and positively genetically correlated with brain gyrification in both infant and adult brains. The polygenic score showed out-of-sample prediction of 3–5.6%, confirmed as largely due to direct effects in sib-pair analyses, and was separately associated with volume of neonatal brain structures involved in motor control. This study offers biological insights into a key behavioural marker of neurodevelopment.

In early childhood, the onset of walking is used as a simple yet robust clinical marker for brain and behavioural development. A major advantage of this milestone is that it is both memorable and clearly defined and therefore can be reliably identified and recalled by parents[1]. Moreover, while there is variability in the sequence and presence of some motor skills (for example, some children bottom shuffle but never crawl), walking is an exclusive and informative milestone for both typical and atypical development.

In current clinical practice, an inability to walk independently by age 18 months is used in national guidelines such as those outlined by the UK National Institute of Health and Care Excellence (NICE; https://www.nice.org.uk/) or by the US Centers for Disease Control and Prevention[2] as a screening criterion for referral to a paediatrician for further assessment and investigation[3]. This is because delayed walking could represent an underlying motor-specific issue such as a primary muscle disorder or generalized issues such as global developmental delay[4]. The causes of these issues can be genetic or environmental, including genetic disorders and extreme prematurity[5]. However, historical data suggest that only a minority (about a third) of late walkers may have an underlying neurological abnormality or developmental disorder, and

**Table 1 | Genome-wide significant loci associated with age at onset of walking**

| Genomic locus | Lead SNPs | Chromosome | Position | A1 | A2 | A1 freq. in EUR | N | Effect size | s.e. | P | COJO P | Nearest genes |
|---|---|---|---|---|---|---|---|---|---|---|---|---|
| 1 | rs7956202 | 12 | 112661263 | T | G | 0.831 | 64,273 | 0.098 | 0.015 | $2.045×10^{-11}$ | $1.856×10^{-11}$ | *HECTD4* |
| 2 | rs16952251 | 16 | 53483138 | A | G | 0.697 | 64,286 | −0.082 | 0.012 | $2.637×10^{-11}$ | $2.470×10^{-11}$ | *RBL2* |
| 3 | rs73030207 | 5 | 1902324 | A | C | 0.014 | 64,266 | 0.230 | 0.040 | $5.454×10^{-9}$ | $5.691×10^{-11}$ | *CTD-2194D22.4* |
| 4 | rs28383314 | 6 | 32587213 | T | C | 0.339 | 60,831 | −0.078 | 0.012 | $1.028×10^{-10}$ | $9.863×10^{-11}$ | *HLA-DQA1* |
| 5 | rs10010217 | 4 | 80801911 | T | C | 0.718 | 70,313 | 0.081 | 0.013 | $4.097×10^{-10}$ | $4.698×10^{-10}$ | *PCAT4, ANTXR2* |
| 6 | rs382362 | 17 | 43691377 | T | C | 0.758 | 59,830 | −0.098 | 0.016 | $5.370×10^{-10}$ | $5.209×10^{-10}$ | *RPS26P8* |
| 7 | rs4785475 | 16 | 50939789 | A | G | 0.277 | 64,263 | 0.081 | 0.013 | $1.385×10^{-9}$ | $1.439×10^{-9}$ | *RP11-883G14.1* |
| 8 | rs148420384 | 13 | 31826394 | C | G | 0.668 | 58,121 | −0.077 | 0.013 | $2.341×10^{-9}$ | $2.414×10^{-9}$ | *B3GALTL* |
| 9 | rs1559625 | 2 | 60173866 | A | G | 0.390 | 60,838 | 0.068 | 0.012 | $2.329×10^{-8}$ | $2.564×10^{-8}$ | *RP11-444A22.1* |
| 10 | rs6058302 | 20 | 34290037 | T | C | 0.140 | 60,884 | −0.099 | 0.018 | $4.188×10^{-8}$ | $4.481×10^{-8}$ | *ROMO1, RBM39* |
| 11 | rs11958405 | 5 | 22247159 | A | G | 0.515 | 70,536 | 0.060 | 0.011 | $5.289×10^{-8}$ | $4.810×10^{-8}$ | *CDH12* |

The allele frequency in the 1000 Genomes[85] European-ancestry sample (EUR), the effect sizes and the standard errors (s.e.) refer to Allele 1 (A1). The P values of association from the meta-analysis performed in METAL and the P values resulting from the conditional and joint (COJO)[29] analysis are reported. The nearest genes were identified using FUMA[33].

that variation in age at onset of walking within the typical range might not be strongly associated with IQ in childhood[6]. As such, late-walking children (later than 18 months) might either reflect an extreme of typical variation or relate to clinically meaningful conditions with a later age of onset.

Although most humans begin to walk independently by early childhood, typical attainment of this milestone can be achieved within a relatively wide developmental period, for most infants between 8 and 18 months old[3]. It is thought that age at onset of independent walking (hereafter, AOW) is a complex trait determined by multiple factors, including body dimensions, year of birth, gestational age and related neural maturation, opportunity to practice[7,8], cultural context[9] and nutrition[10]. Many of these factors are thought to influence the structure and function of a network of brain areas implicated in motor control, including the cortex, basal ganglia and cerebellum, with dysfunction in these brain regions resulting in movement disorders[11]. In addition to reflecting general developmental processes, the ability to walk independently may itself have cascading effects on other developmental domains[12]. When children transition from crawling to standing and walking, the perspective at which they perceive the world changes, as do their means of interacting with the world[13]. However, it remains unclear what are the causal influences underlying the wide variability in age at onset of walking or whether these causal influences are also associated with later health, neurodevelopmental and cognitive outcomes.

A greater understanding of the variability and causes of late walking has clear societal implications. It would inform many countries' public health policy that aim to screen children for delay[14]. Genetic information has the potential to offer greater understanding regarding the aetiology of this developmental milestone. Furthermore, it can contribute alongside screening tools to aid the prediction and early identification of clinically relevant conditions associated with early or delayed onset of walking, and avoid missing time for potentially beneficial physical training when appropriate.

There is substantial evidence for a genetic contribution to motor development. A recent meta-analysis of infant twin studies showed that the broad category of psychomotor function was one of the most heritable behavioural domains, with pooled heritability of 59%[15]. For AOW specifically, a study of 2,274 twin pairs in England and Wales reported a heritability of 84%[16]. Polygenic scores for autism spectrum disorder (ASD, hereafter autism), schizophrenia and bipolar disorder have been found to be associated with infant neuromotor characteristics such as muscle tone, reflexes and senses[17]. Further, the attention deficit/hyperactivity disorder (ADHD) polygenic score was associated with AOW[18].

As such, age at onset of walking appears to be an ideal candidate for genetic discovery research. Identification of specific genetic loci is an important step towards uncovering the biological mechanisms underlying this developmental milestone and deriving clinically informative insights with respect to childhood motor disorders. There have been no common gene discovery studies of AOW so far[19].

In sum, there are several reasons for focusing on AOW. It is a marker of brain and behavioural development, it is easily measurable in large cohorts, reliably recalled by parents[1] and varies substantially between children.

Here we present a genome-wide association study (GWAS) meta-analysis of AOW in a sample of 70,560 children from four European-ancestry cohorts. First, we aimed to quantify single-nucleotide polymorphism (SNP)-based heritability of AOW and the degree of polygenicity of this trait. Second, we aimed to identify independent genetic loci associated with AOW and their functional roles. Third, we estimated genetic correlations with physical health indicators, cognitive traits, neurodevelopmental conditions, psychiatric disorders and cortical phenotypes. Fourth, we evaluated the predictive power of the AOW polygenic score and tested whether it was associated with the volume of neonatal brain structures in an independent cohort.

## Results

### Genomic loci associated with age at onset of walking

We conducted a GWAS meta-analysis of AOW in 70,560 children including data from four European-ancestry cohorts: the Norwegian Mother, Father and Child Cohort Study[20,21] (MoBa, N = 58,302), the Netherlands Twin Register[22] (NTR, N = 6,251), the Lifelines multigenerational prospective population-based birth cohort study[23] from the North of the Netherlands (N = 3,415) and the United Kingdom Medical Research Council National Study for Health and Development[24] (NSHD, N = 2,592). Analyses were preregistered on OSF (https://doi.org/10.17605/OSF.IO/M2QV3). The quantile–quantile (QQ) plot for the MoBa GWAS (Supplementary Fig. 2) indicated a P value deviation from a normal distribution ($\lambda_{GC}$ = 1.227). The observed inflation is probably explained by trait polygenicity (linkage disequilibrium score regression [LDSC] intercept = 1.008 (0.008)[25,26]; see Supplementary Note A for a detailed investigation of the observed inflation). The other smaller cohorts' inflation factors were below the recommended threshold of 1.10 (NTR $\lambda_{GC}$ = 0.975, Supplementary Fig. 4; Lifelines $\lambda_{GC}$ = 1.001, Supplementary Fig. 6; NSHD $\lambda_{GC}$ = 1.002, Supplementary Fig. 8), which is expected given the positive relationship between inflation and sample size[27]. Therefore, contrary to the preregistered plan, automatic correction

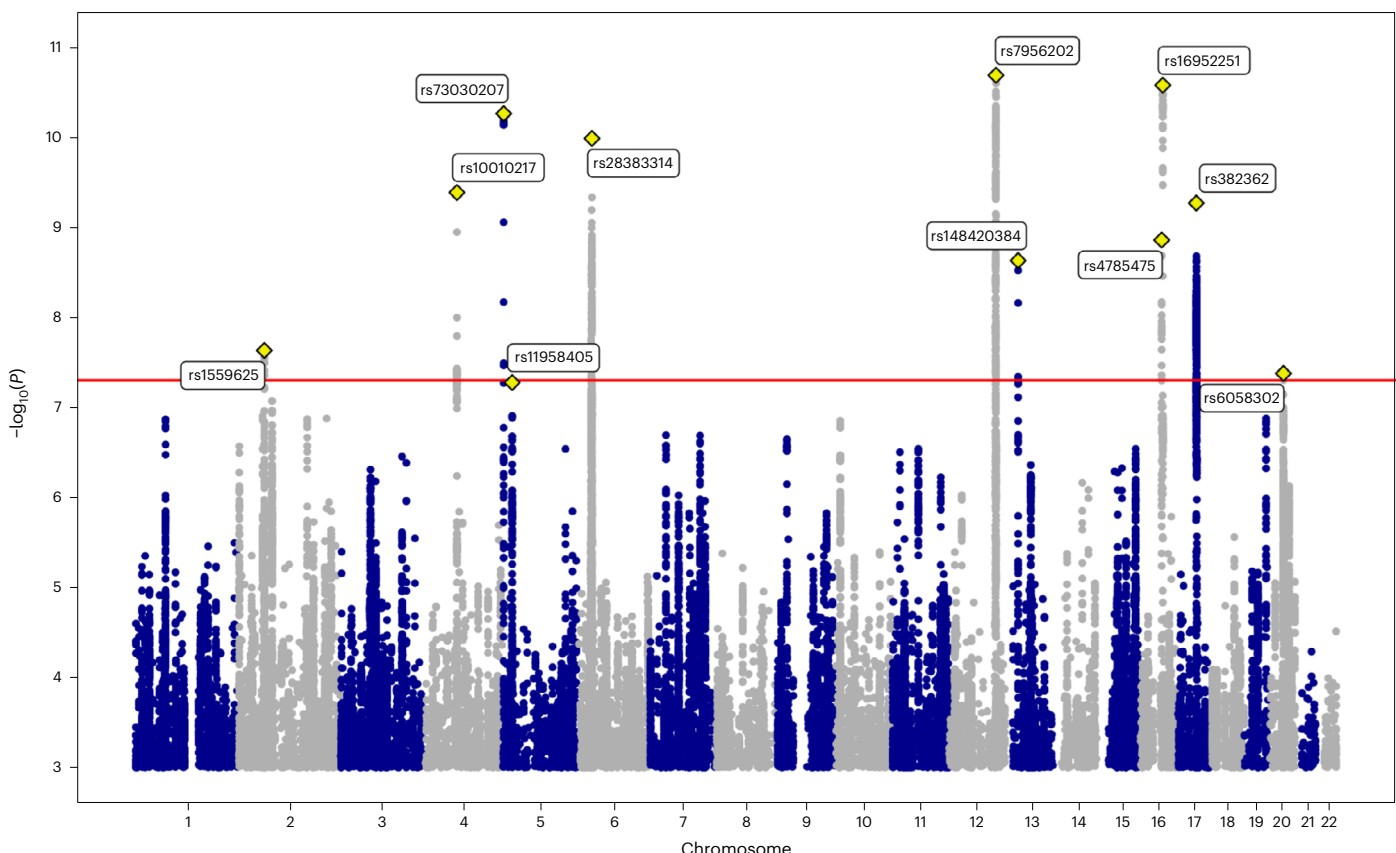

**Fig. 1 | Manhattan plot of the GWAS meta-analysis of age at onset of walking.**
The *x* axis shows genomic position (chromosomes 1–22) and the *y* axis shows statistical significance as $-\log_{10}(P \text{value})$. *P* values are two-sided and based on an inverse-variance standard-error-weighted fixed-effects meta-analysis. *N* = 70,560. The horizontal red line indicates the *P*-value threshold for genome-wide statistical significance ($P = 5 \times 10^{-8}$). *P* values were not adjusted for multiple comparisons.

The lead SNP for each genome-wide significant locus is labelled and indicated with a yellow diamond. The inflation factor $\lambda_{GC}$ for this GWAS was 1.27 and LDSC intercept was 1.00 (s.e. = 0.01), suggesting that inflation was due to polygenicity of AOW (see Supplementary Note A for a discussion). The meta-GWAS QQ plot by allele frequency is presented in Supplementary Fig. 9. SNPs with *P*-values < 0.001 (corresponding to $-\log_{10}(P) > 3$) are presented as data points.

for genomic control was not applied for all cohorts when performing the standard-error-weighted meta-analysis using the METAL tool[28].

We identified 2,525 genome-wide significant SNPs ($P < 5 \times 10^{-8}$), of which 11 were independent loci with one lead variant per locus in GCTA conditional and joint analysis (COJO)[29] (Table 1 and Fig. 1, see also Supplementary Fig. 9 for the QQ plot and Supplementary Fig. 10 for the regional plots). All 11 lead SNPs remained significant after conditioning on the other significant SNPs on the same chromosome (Table 1, column 'COJO P'). The most strongly associated SNP was located on chromosome 12 (rs7956202 near *HECTD4*, $P = 2.045 \times 10^{-11}$). The second most significant lead SNP was located on chromosome 16 (rs16952251, near *RBL2*, $P = 2.637 \times 10^{-11}$) (fine mapping of this locus is discussed later; see Results section 'Co-localization with gene expression in the brain'). See Table 1 for a full list of significant loci, Supplementary Table 4 for previous associations with complex traits, and Supplementary Table 5 for which cohorts contributed to each locus.

### Common genetic architecture of age at onset of walking
SNP-based heritability of AOW estimated with LDSC[25] was $h^2_{SNP} = 24.13\%$ (95% CI = 21.856, 26.404). Heritability for the phenotype in males (*N* = 35,642) and females (*N* = 34,918) was estimated to be 23.06% (95% CI = 19.512, 26.608) and 23.06% (95% CI = 19.356, 26.764), respectively. The genetic correlation ($r_g$) of the phenotype between males and females estimated with LDSC[30] was 0.99 (95% CI = 0.872, 1.108).

The SNP-based heritability ($h^2_{SNP}$) estimated using LDSC[25] for the MoBa sample was $h^2_{SNP} = 25.11\%$ (95% CI = 22.484, 27.736) and for the NTR sample, $h^2_{SNP} = 19.09\%$ (95% CI = 4.547, 33.633). Lower $h^2_{SNP}$

estimates and larger standard errors were obtained for the smaller samples, namely: Lifelines ($h^2_{SNP} = 9.52\%$, 95% CI = −15.921, 34.961) and NSHD ($h^2_{SNP} = -3.02\%$, 95% CI = −36.673, 30.633), as LDSC cannot produce reliable estimates with samples <5,000 (ref. 25). Genetic correlation between MoBa and NTR was $r_g = 0.893$ (95% CI = 0.558, 1.228, $P = 1.803 \times 10^{-7}$) and between NTR and Lifelines, $r_g = 0.463$ (95% CI = −0.623, 1.549, $P = 0.404$). As expected, other genetic correlations were out of bound (MoBa–Lifelines $r_g = 1.168$, 95% CI = −0.233, 2.569, $P = 0.103$) or non-estimable due to low reliability of the LDSC estimates, indicated by the large SNP-based heritability standard errors obtained for the smaller cohorts. Of note, the interval between AOW and parent report was not significantly correlated with the mean AOW difference between cohorts ($r = 0.16$, $P = 0.76$, two-tailed).

There was no genome-wide statistically significant heterogeneity (using the conventional $P < 5 \times 10^{-8}$ threshold) between cohorts as tested with the heterogeneity metric per SNP, $I^2$; the maximum $I^2$ was 95.3 for SNPs rs7864115 ($\chi^2_{(1)} = 21.453$, $P = 3.627 \times 10^{-6}$) and rs148684045 ($\chi^2_{(1)} = 21.441$, $P = 3.648 \times 10^{-6}$). This indicates that variation of individual SNP effects between individual GWASs was not due to heterogeneity between the cohorts[31] (Supplementary Fig. 11). Overall, the *M* multiSNP heterogeneity metric across the independent lead SNPs[32] associated with AOW indicated no systematically more or less influential study (see Supplementary Table 3, all Bonferroni-corrected *P*s < 0.401).

### Biological annotation of associated loci and genes
**Analyses on prioritized genes annotated to significant SNPs.** The genome-wide significant SNPs were mapped to 233 genes on the

basis of genomic position, expression quantitative trait loci (eQTLs) and chromatin interaction information in FUMA[33] (Supplementary Table 6). We tested whether these prioritized genes were differentially expressed in the brain across BrainSpan[34] developmental stages and GTEx (v.8)[35] tissues. We observed a significant downregulation of the differentially expressed genes (DEGs) in multiple tissues including the brain (amygdala and hippocampus, both-sided DEG Bonferroni-adjusted $P = 5.979 \times 10^{-8}$, $P = 5.165 \times 10^{-7}$) and the heart left ventricle ($P_{adj.} = 3.124 \times 10^{-7}$), and DEGs upregulation in fibroblasts ($P_{adj.} = 0.003$) (Supplementary Fig. 12). The enrichment of upregulated or downregulated DEGs across BrainSpan developmental stages was not significant (all Bonferroni-adjusted $P = 1$, see Supplementary Table 7 for full results). Gene sets associated with AOW were enriched in the Gene Ontology[36] neurogenesis and generation of neurons pathways (see Supplementary Table 8 for all significantly enriched gene sets and gene set–trait associations from previous studies).

**Genes associated with age at onset of walking.** The MAGMA[37] gene-based test performed in FUMA on the meta-GWAS summary statistics indicated 50 genes that were associated with AOW at a Bonferroni-corrected genome-wide significance threshold of $2.664 \times 10^{-6}$ ($P = 0.05$ divided by 18,766 genes; Supplementary Table 9). A full list of previously reported genome-wide associations with complex traits for the 50 AOW-associated genes is provided in Supplementary Table 10.

Using the Genomics England PanelApp[38], we found that 13 (27.7%) of the 47 of the 50 MAGMA genes that had Ensembl IDs in PanelApp were associated with intellectual disability (ID, v.5.557); this is over double the proportion (2.10 times) of ID-associated genes in the panels as a whole (2,624 out of 19,950, 13.2%; $\chi^2_{(1)} = 7.45$; $P = 0.006$, two-tailed). These genes include *ATXN2*, *AUTS2*, *CUX2*, *FOXP1*, *KANSL1* and *RBL2* (Supplementary Table 9). Furthermore, we found that 7 of the 47 genes were associated with autism (14.9%), which is over 4 times the proportion of autism-associated genes in the panel (v.0.36, largely based on SFARI gene[39]) as a whole (734 out of 19,950, 3.68%; $\chi^2_{(1)} = 13.7$; $P = 0.0002$, two-tailed).

To identify tissue specificity of AOW, MAGMA gene-property analyses performed in FUMA using gene-based association $P$ values for all the 18,766 genes revealed that gene expression was primarily enriched in the brain cerebellar hemispheres ($\beta = 0.017$, 95% CI = −0.050, 0.084, $P = 0.006$) and cerebellum ($\beta = 0.018$, 95% CI = −0.052, 0.088, $P = 0.007$), although these results were not significant at a Bonferroni-corrected $\alpha$ level of 0.05 for 54 tissues ($9.000 \times 10^{-4}$; see Supplementary Fig. 13). Overall, expression of the genes associated with AOW was significantly enriched between 19 and 24 post-conceptional weeks (late mid-prenatal period, $\beta = 0.041$, 95% CI = 0.011, 0.070, $P = 0.004$; Supplementary Fig. 14). The MAGMA gene-set analysis yielded no significant results (Supplementary Table 11).

**Analyses on the meta-GWAS summary statistics.** Enrichment of AOW meta-GWAS signal by functional genomic annotation was tested using stratified LDSC[40] analyses. These revealed that heritability of AOW was significantly enriched in genomic regions conserved in primates (16.142-fold enrichment, 95% CI = 10.421, 21.863, $P = 0.309 \times 10^{-6}$), mammals (13.053-fold enrichment, 95% CI = 8.239, 17.867, $P = 0.287 \times 10^{-5}$,) and vertebrates (8.747-fold enrichment, 95% CI = 5.450, 12.044, $P = 0.817 \times 10^{-5}$; see Extended Data Fig. 1). Full results of partitioned heritability by functional genomic annotation can be found in Supplementary Table 12.

We then tested whether heritability was enriched in specific cell types using stratified LDSC[41] and found significant enrichment in the brain, particularly in the basal ganglia (caudate: enrichment = $1.400 \times 10^{-8}$, Bonferroni-adjusted $P = 0.014$, 95% CI = $6.062 \times 10^{-9}$, $2.194 \times 10^{-8}$, nucleus accumbens: enrichment = $1.760 \times 10^{-8}$, $P_{adj.} = 0.001$, 95% CI = $9.740 \times 10^{-9}$, $2.546 \times 10^{-8}$, putamen: enrichment = $1.470 \times 10^{-8}$,

$P_{adj.} = 0.006$, 95% CI = $6.840 \times 10^{-9}$, $2.256 \times 10^{-8}$), cortex (enrichment = $1.370 \times 10^{-8}$, $P_{adj.} = 0.003$, 95% CI = $6.781 \times 10^{-9}$, $2.062 \times 10^{-8}$), amygdala (enrichment = $1.360 \times 10^{-8}$, $P_{adj.} = 0.020$, 95% CI = $5.682 \times 10^{-9}$, $2.152 \times 10^{-8}$) and cerebellum (enrichment = $1.320 \times 10^{-8}$, $P_{adj.} = 0.014$, 95% CI = $5.772 \times 10^{-9}$, $2.023 \times 10^{-8}$; Extended Data Fig. 2). Complete stratified LDSCs by cell-type estimate are reported in Supplementary Table 13.

**Co-localization with gene expression in the brain**

We investigated whether genes near the 11 genome-wide significant loci, as well as 50 genes significantly associated with AOW (Supplementary Table 9), were enriched for eQTLs in an independent dataset of post-mortem bulk RNA-seq from 261 samples of the human adult cerebellum[42]. We identified significant eQTLs for the gene *RBL2* (which encodes a transcriptional regulator by the same name) in genomic locus 2 on chromosome 16 (Table 1). Comparing the statistical evidence of association with AOW (GWAS) against the statistical evidence of association with *RBL2* expression, we noticed a distinct pattern: both the GWAS and eQTL $P$ values had two groups of significantly associated SNPs distinguished by their linkage disequilibrium correlation with a lead GWAS SNP (rs17800727, Fig. 2a). Group 1 had the strongest evidence for GWAS association (min $P = 2.95 \times 10^{-11}$) but slightly weaker evidence of eQTL association (min $P = 2.72 \times 10^{-13}$ cerebellum eQTL), while Group 2 had weaker evidence for GWAS association (min $P = 9.51 \times 10^{-8}$) but stronger evidence of eQTL association (min $P = 6.41 \times 10^{-24}$ cerebellum eQTL, Fig. 2a). We investigated the probability that the same SNPs in this locus influence both AOW and *RBL2* expression (co-localization, Fig. 2). Our co-localization analysis at this locus suggested an independent causal variant in the GWAS (rs17800727; chr16:53481010:A:G GRCh37; chr16:53447098:A:G GRCh38) and the eQTL data (rs7203132; chr16:53429775:G:A GRCh37; chr16:53395863:G:A GRCh38) with a posterior probability (PP) of 0.96 (ref. 43) that the causal SNP is distinct in each dataset. A similar co-localization pattern was observed using 1,433 samples of the human adult cortex (ref. 42) (Supplementary Note B and Fig. 15; PP = 0.97–0.99).

To understand these two groups, we assessed their distribution across the 2-Mb genomic locus (±1 MB around the gene) and observed that they overlapped throughout a 125-kb peak with well-defined margins for both the GWAS and *RBL2* eQTL analysis (Fig. 2b). We next considered how these SNPs were distributed on the basis of minor allele frequency (MAF, Fig. 2c). The Group 1 SNPs (strongest GWAS evidence) had a MAF of 30%, while the Group 2 SNPs (strongest eQTL evidence) had a MAF of 50%. Using whole-genome sequencing data from 176 individuals with paired post-mortem RNA-seq data from prefrontal cortex[44], we used the MAF distribution to identify five haplotypes (Fig. 2d) and each individual's genotype. Group 2 SNPs (strongest eQTL evidence, MAF 50%) are found in three haplotypes (dark blue and red, dark blue and yellow, dark blue alone, Fig. 2d) resulting in the high MAF of 50%. Homozygous status for the Group 2 SNPs is associated with decreased expression of *RBL2* (Wilcoxon rank test, two-sided; $W_{(56)} = 249$, $P = 0.007$, Hodges–Lehmann estimator = −2.105, 95% CI = −3.813, −0.610). We infer that one of the SNPs shown in dark blue (Fig. 2c) impacts *RBL2* expression, although no clear candidate SNP was evident when considering epigenetic data.

Group 1 SNPs are only found on one haplotype (dark blue and red, Fig. 2d) resulting in a lower MAF of 30% than the Group 2 SNPs. We infer that one of the Group 1 SNPs has a functional impact above and beyond the decrease in *RBL2* expression mediated by the Group 2 SNPs, to yield the stronger evidence of association with AOW. Annotation of the 125-kb locus with VEP[45] identified rs17800727 as a likely candidate for this effect, since it results in a missense variant (MANE isoform: ENST00000262133.11, p.Tyr210Cys) (Fig. 2e) that is predicted to impact function by some severity metrics (for example, 'Damaging' based on PolyPhen2 (ref. 46), CADD[47] score of 25) but not all (for example, 'Tolerated' based on SIFT). If the missense variant had a loss-of-function effect, it would be on a haplotype that magnifies the functional impact

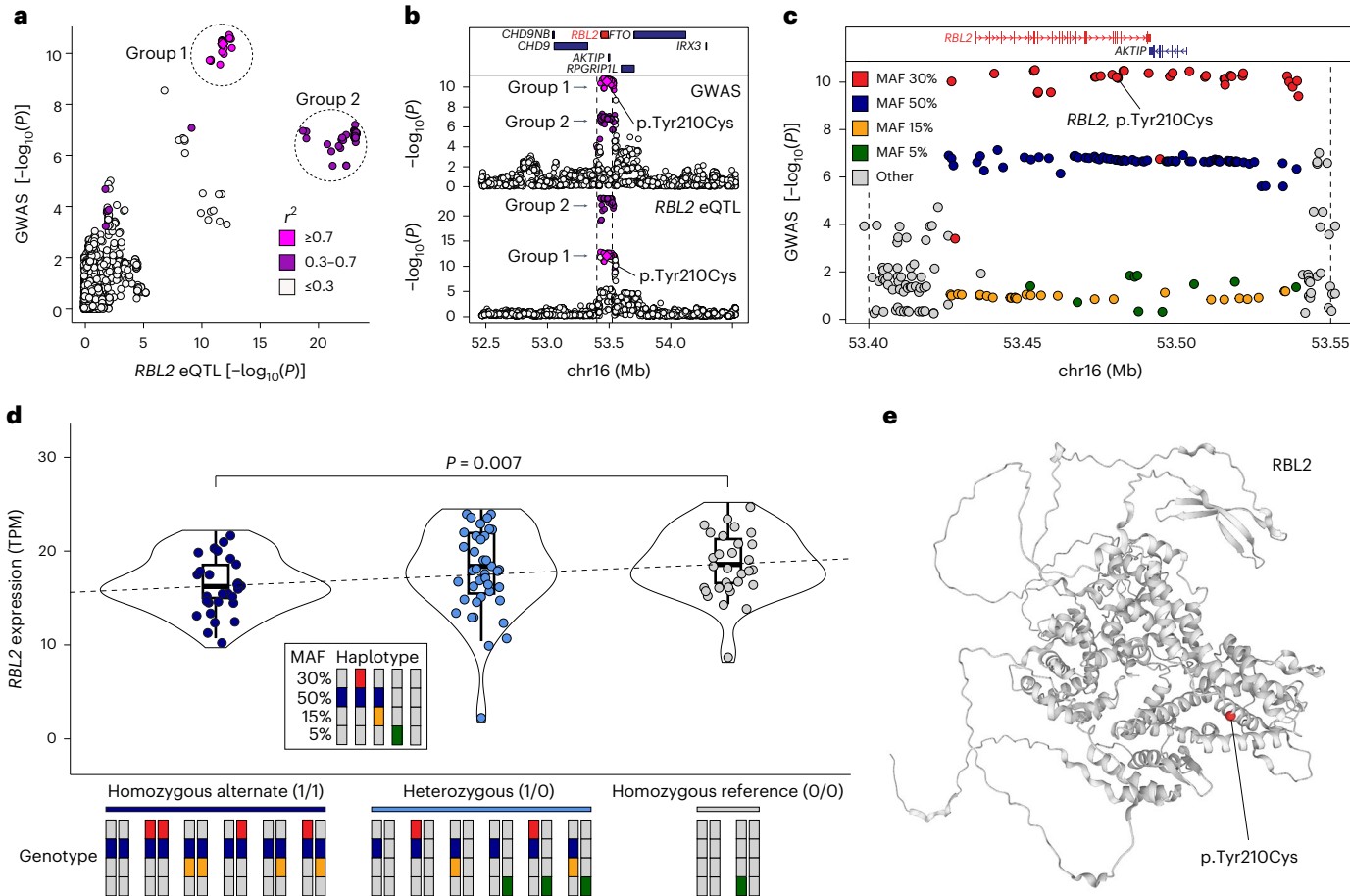

**Fig. 2 | Co-localization of variants in genomic locus 2.** Genomic locus 2 overlaps with a region in which SNPs are predicted to alter *RBL2* expression in the human brain (eQTLs). **a**, The GWAS evidence for association with age at onset of walking [$-\log_{10}(P\text{value})$, *y* axis] is plotted against the statistical evidence of being an eQTL for *RBL2* in human adult cerebellum[42] [$-\log_{10}(P\text{value})$, *x* axis] for each SNP (points) within a 2-Mb window around the GWAS peak. Points are coloured by linkage disequilibrium (LD) correlation with the lead SNP (rs17800727) and these values were used to define two groups. **b**, The SNPs from **a** are shown in the 2-Mbp genomic region (*x* axis, GRCh37) with protein-coding genes (top), GWAS evidence for association with age at onset [$-\log_{10}(P\text{value})$, middle] and statistical evidence for *RBL2* expression in human cerebellum [$-\log_{10}(P\text{value})$, *y* axis, bottom]. Point colour matches **a**. **c**, A zoomed-in view of the peak indicated by dashed vertical lines in **b** shows the GWAS evidence for association with age at onset of walking [$-\log_{10}(P\text{value})$, *y* axis] by genomic position (*x* axis, GRCh37). Colour indicates the MAF of each SNP. The locations of protein-coding genes in

the region are indicated at the top. An SNP (rs17800727) that results in a missense variant (p.Tyr210Cys) in *RBL2* is marked. **d**, Swarm, violin and boxplots showing the distribution of *RBL2* expression in the prefrontal cortex (transcripts per million (TPM), *y* axis). Each point represents the expression of *RBL2* in 1 of 87 prenatal human cortices (BrainVar[44]) split by genotype into 3 groups on the basis of zygosity for the Group 2 50% MAF SNPs. The *P* value represents the difference between the homozygous alternate (*N* = 28) and homozygous reference (*N* = 30) groups. The centre is the median expression value. The lower and upper bounds of the box correspond to the first and third quartiles (the 25th and 75th percentiles). The upper/lower whiskers extend from the upper/lower bound to the largest/smallest value no further than 1.5× the interquartile range. Data beyond the end of the whiskers are outlying points and are plotted individually. Bars at the bottom indicate pairs of haplotypes (derived from the data shown in **c** making up each genotype). **e**, Structure of the RBL2 protein predicted by AlphaFold[95] with the location of rs17800727, p.Tyr210Cys in red[96].

through decreased expression of *RBL2*; future functional studies would be required to validate this impact.

We also identified co-localization of SNPs associated with expression of several genes in both the cerebellum and cortex with SNPs associated with AOW in genomic locus 6 on chromosome 17 (Table 1). This region has a complex haplotype structure, including alternative contigs, which may explain this result. In cerebellum, we identified co-localization in *KANSL1* (PP = 0.79), *PLEKHM1* (PP = 0.78), *SPPL2C* (PP = 0.77) and *STH* (PP = 0.63). In the cortex, we also identified co-localization in *STH* (PP = 0.78) and *SPPL2C* (PP = 0.72), as well as in *CRHR1* (PP = 0.74).

**Polygenic score analysis**

In a leave-one-out design, we calculated a polygenic score (PGS) on the basis of meta-analyses of all samples, leaving out either Lifelines, NTR or NSHD. In the Lifelines cohort, the PGS from the meta-GWAS of the

other cohorts (MoBa, NTR and NSHD) was significantly associated with AOW ($\beta$ = 0.185, 95% CI = 0.152, 0.217, $P < 2 \times 10^{-16}$, $R^2$ = 0.034). Using the same method, the PGS was significantly associated with AOW in the NTR cohort ($\beta$ = 0.185, 95% CI = 0.147, 0.223, $P < 2 \times 10^{-16}$, $R^2$ = 0.031) and in the NSHD cohort ($\beta$ = 0.175, 95% CI = 0.137, 0.213, $P < 2 \times 10^{-16}$, $R^2$ = 0.030). The MoBa sample comprised a high proportion of the data such that it would be inappropriate as a 'left out' sample in a leave-one-out design. Therefore, we applied 5-fold cross-validation to this cohort, yielding 5 within-sample PGSs with a mean variance explained of $R^2$ = 0.056 (s.e. = 0.001).

Genetic effects identified by GWAS can be confounded by indirect genetic effects, for example, through population structure, assortative mating and passive gene–environment correlation (prGE)[48]. To identify possible confounding from indirect genetic effects, we used a within- and between-sib-pair PGS analysis. We generated a PGS from a meta-analysis of the MoBa, Lifelines and NSHD GWAS summary

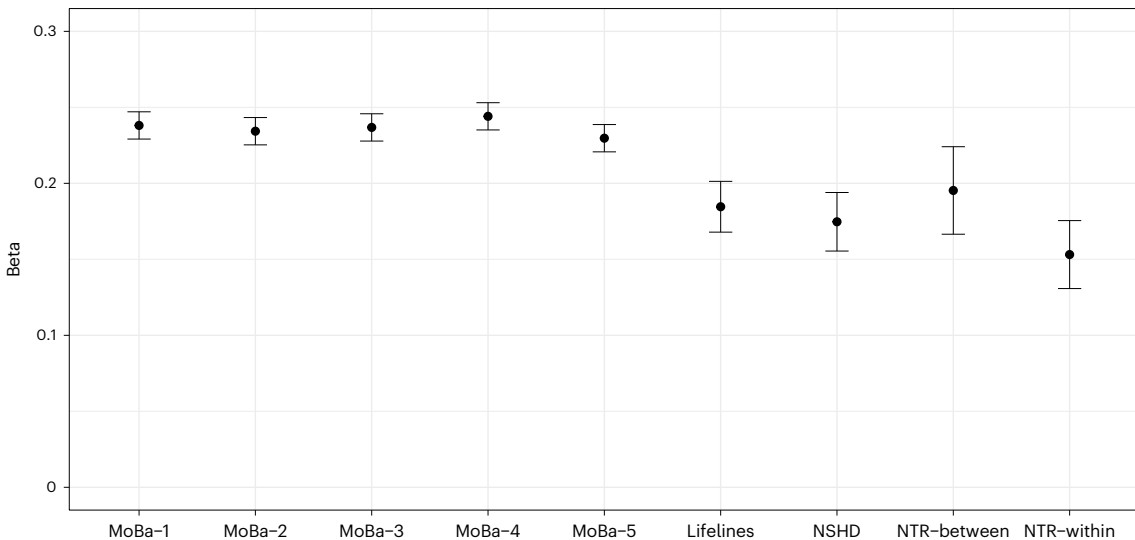

**Fig. 3 | Beta estimates of the prediction of age at onset of walking for the five MoBa subsamples, Lifelines, NSHD, NTR between- and NTR within-sib-pair polygenic score analyses.** Data are presented as beta estimates ±s.e. of the beta estimate of a linear regression model testing the association between age at onset of walking and the polygenic score (two-tailed $P$ values). $N = 11,660$ (MoBa-1, MoBa-2, Moba-3), $N = 11,661$ (MoBa-4, MoBa-5), $N = 3,415$ (Lifelines), $N = 2,592$ (NSHD); $N = 2,508$, $N$ pairs = 1,254 (NTR between- and NTR within-sib-pair).

statistics and used it to conduct within-family associations in the NTR dataset. Among 1,254 dizygotic twin pairs ($N = 2,508$ individuals), within- and between-family standardized regression coefficients in a linear mixed-effects model were not significantly different from each other ($\chi^2_{(1)} = 1.479$, $P = 0.224$, two-tailed), indicating that the genetic signal is not biased by prGE, or effects such as stratification and assortative mating. Figure 3 shows the beta estimates of the AOW PGS prediction in all the cohorts, with the NTR within- and between-sib-pair estimates presented separately.

**Genetic correlations with other traits**

Next, we tested for genetic correlations between AOW and a preregistered selection of physical health, neurodevelopmental, psychiatric, cognitive and cortical phenotypes. For physical health, AOW was negatively genetically correlated with childhood body-mass index (cBMI)[49] ($r_g = -0.143$, 95% CI = $-0.217$, $-0.069$, $P = 1.553 \times 10^{-4}$, Bonferroni-adjusted $P = 0.004$) and adult BMI[50] ($r_g = -0.103$, 95% CI = $-0.142$, $-0.063$, $P = 2.858 \times 10^{-7}$, $P_{adj.} = 8.00 \times 10^{-6}$) but not with birth weight ($r_g = 0.068$, 95% CI = $-0.067$, 0.202, $P = 0.325$). Of the six included psychiatric disorders, ADHD[51] showed a significant genetic correlation with AOW ($r_g = -0.180$, 95% CI = $-0.242$, $-0.118$, $P = 1.299 \times 10^{-8}$, $P_{adj.} = 3.64 \times 10^{-7}$). In addition, AOW was positively genetically correlated with the cognitive phenotypes, educational attainment[52] ($r_g = 0.119$, 95% CI = 0.081, 0.157, $P = 7.457 \times 10^{-10}$, $P_{adj.} = 2.088 \times 10^{-8}$) and cognitive performance[53] ($r_g = 0.092$, 95% CI = 0.041, 0.142, $P = 3.967 \times 10^{-4}$, $P_{adj.} = 0.011$).

Among 13 adolescent and adult cortical phenotypes[54], we observed a significant genetic correlation between AOW and folding index ($r_g = 0.136$, 95% CI = 0.062, 0.209, $P = 3.000 \times 10^{-4}$, $P_{adj.} = 0.008$). There were no significant genetic correlations with the other complex traits tested after correction for multiple testing (see Supplementary Table 14 and Fig. 4a). For motor phenotypes, non-preregistered exploratory analyses showed that AOW was genetically correlated with self-reported walking pace in adults[55] ($r_g = 0.058$, 95% CI = 0.006, 0.110, $P = 0.029$, $P_{adj.} = 0.820$), although this result did not survive $P$-value correction for multiple testing (Supplementary Table 14).

In light of our findings of a Bonferroni-significant genetic correlation between AOW and global folding index, we conducted further non-preregistered analyses, as requested by a reviewer, to gain more specific information about the brain regions implicated. We included

regions involved in motor and/or somatosensory function and corrected for multiple testing using false discovery rate (FDR) correction. We found that later AOW was significantly genetically correlated with increased folding in the primary somatosensory cortex (regions of interest (ROIs) in Glasser parcellation[56] 1: $r_g = 0.160$, 95% CI = 0.078, 0.242, FDR-adjusted $P = 0.003$ and 5 m: $r_g = 0.182$, 95% CI = 0.081, 0.283, $P_{adj.} = 0.005$), premotor cortex (ROI 6r: $r_g = 0.152$, 95% CI = 0.042, 0.262, $P_{adj.} = 0.045$) and cingulate motor area (ROI 24dd: $r_g = 0.148$, 95% CI = 0.053, 0.243, $P_{adj.} = 0.021$). See Supplementary Table 16 for the full set of results.

The largest-magnitude genetic correlation was between AOW and ADHD. In light of the potential implications of this finding, we tested, in an exploratory non-preregistered analysis, whether the AOW–ADHD genetic correlation remained after controlling the genetic influences of educational attainment, since the latter are also known to be associated with ADHD[57]. In a genetic multivariable regression performed with GenomicSEM[58], we observed that the relationship between the genetic components of ADHD and AOW remained significant after conditioning for educational attainment (standardized $\beta = -0.160$, 95% CI = $-0.248$, $-0.072$, $P = 3.8 \times 10^{-4}$), while the conditional standardized association between educational attainment and AOW was non-significant ($\beta = 0.038$, 95% CI = $-0.027$, 0.103, $P = 0.246$; Supplementary Fig. 16).

We applied MiXeR univariate and bivariate Gaussian mixture modelling[59], which calculates the polygenicity of AOW defined as the number of SNPs that explain 90% of the $h^2_{SNP}$, and the genetic overlap between AOW and other phenotypes, including SNPs of both concordant and discordant effect directions. We applied bivariate mixture modelling to AOW with all other phenotypes with which there was a significant genetic correlation as calculated by LDSC after correction for multiple testing (based on Fig. 4a). In terms of Akaike information criterion (AIC) fit, we found support for the bivariate MiXeR models that estimated the optimal polygenic overlap between AOW and childhood and adult BMI, educational attainment, cognitive performance, ADHD and folding index (see Fig. 4b, AIC and Bayesian information criterion (BIC) values for all correlated phenotypes are provided in Supplementary Table 15). These models were supported over the 'minimal model' which explains the observed LDSC models using the minimal amount of polygenic overlap possible.

The polygenicity of AOW was 11,857 SNPs, confirming the hypothesis that the inflation observed in the QQ plot could be explained by

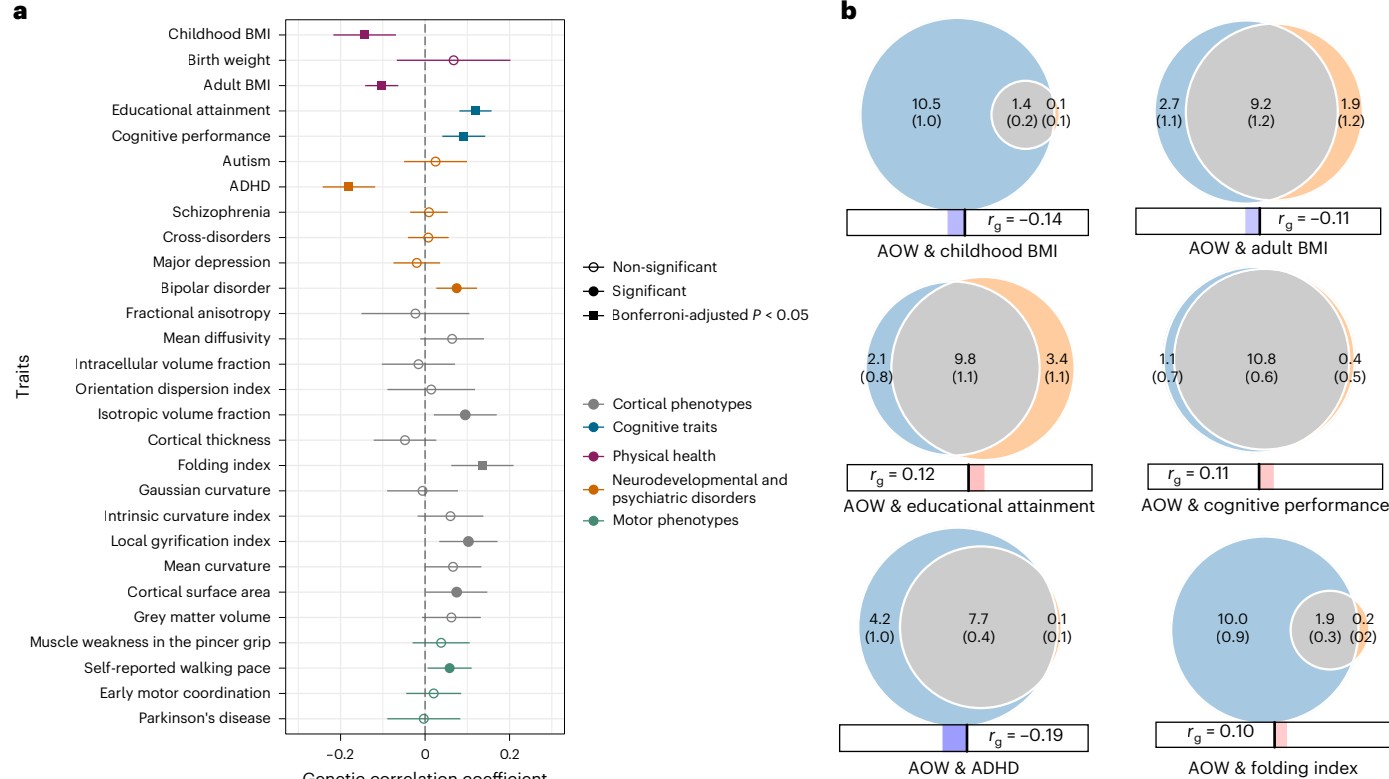

**Fig. 4 | Genetic overlap between age at onset of walking and other complex traits. a**, Genetic correlation between AOW and physical health (purple), cognitive traits (blue), neurodevelopmental conditions and psychiatric disorders (orange), cortical phenotypes (grey) and non-preregistered motor phenotypes (green). Data are presented as correlation coefficients ± 95% CIs. Filled circles indicate significant correlations based on CIs. Filled squares indicate the traits that remain significantly genetically correlated with age at onset of walking after adjusting the two-sided $P$ values obtained from LDSC for multiple testing using Bonferroni correction. The maximum GWAS sample sizes for each of the traits included in the LDSC analysis are as follows: age at onset of walking $N = 70,560$; childhood BMI $N = 61,111$; birth weight $N = 42,212$; adult BMI $N = 795,640$; educational attainment $N = 765,283$; cognitive performance $N = 269,867$; autism $N_{cases} = 18,382$, $N_{controls} = 27,969$; ADHD $N_{cases} = 38,691$, $N_{controls} = 186,843$; schizophrenia $N_{cases} = 67,390$, $N_{controls} = 94,015$; cross-disorders $N_{cases} = 232,964$, $N_{controls} = 494,162$,; major depression $N_{cases} = 170,756$, $N_{controls} = 329,443$; bipolar disorder $N_{cases} = 41,917$, $N_{controls} = 371,549$; cortical phenotypes (fractional

anisotropy, mean diffusivity, intracellular volume fraction, orientation dispersion index, isotropic volume fraction, cortical thickness, folding index, Gaussian curvature, intrinsic curvature index, local gyrification index, mean curvature, cortical surface area, grey matter volume) $N = 36,663$; muscle weakness in the pincer grip $N_{cases} = 48,596$, $N_{controls} = 207,927$; self-reported walking pace $N = 450,967$; early motor coordination $N = 31,797$; Parkinson's disease $N_{cases} = 26,421$, $N_{controls} = 442,271$. **b**, Venn diagrams representing MiXeR bivariate analyses between AOW and the 6 other phenotypes with which it has Bonferroni-significant genetic correlations. The size of the circles and the numbers within them represent the relative polygenicity of each trait (that is, how many genetic variants contribute to 90% of the SNP heritability). The overlap between each pair of circles represents the degree of genetic overlap between the two phenotypes, that is, the number of shared variants in thousands, along with the standard error. Numbers and standard errors in sections of the circles that do not overlap represent the number of variants unique to that phenotype. The corresponding $r_g$, estimated using LDSC, is shown below each Venn diagram.

trait polygenicity (Supplementary Note A). MiXeR presents the genetic overlap between two traits as Venn diagrams (Fig. 4b). In terms of the proportion of the SNPs contributing to the polygenicity of AOW that overlap with other phenotypes investigated, the traits investigated that showed the most overlap were cognitive performance (91.07%), educational attainment (82.44%), adult BMI (77.38%) and ADHD (64.87%). Of these overlapping SNPs, the fractions of SNPs that had concordant directions of effect were 55.10% and 53.71% for educational attainment and cognitive performance, respectively. On the contrary, little SNP overlap, despite significant genetic correlation, was found with childhood BMI (11.80%, of which 36.44% was concordant) and folding index (15.84%, of which 58.72% was concordant). A summary of all bivariate MiXeR analysis results can be found in Supplementary Table 15.

**Polygenic score association with brain measures at birth**
In an exploratory analysis, we tested whether the PGS for AOW was associated with measurable differences in infant brain volume and gyrification at birth. We used neonatal T2 imaging data from a European subsample of 264 term-born infants (137 male, 127 female), acquired as part of the Developing Human Connectome Project (dHCP)[60].

The effect of the AOW PGS on brain volume was investigated across the whole brain at the voxel level using log-Jacobian determinants, calculated using nonlinear deformation fields between participants and the dHCP neonatal standardized atlas. In the resultant maps, higher values represent brain regions that contracted during image registration (that is, had larger brain volumes), while smaller values represent volume reductions[61]. We performed a tensor-based morphometry analysis, applying a general linear model (GLM) and permutation testing for statistical inference. We found a significant positive correlation between the AOW PGS and regional brain volume in the right basal ganglia, right posterior thalamus, bilateral anterior thalami, bilateral cerebellum and cerebellar peduncles, pons, medulla, primary visual cortex and superior temporal sulcus after correcting for multiple comparisons and thresholding at a corrected $P < 0.05$ (Fig. 5). Increased brain volume in these regions was associated with a higher PGS (predisposing to later AOW).

To explore whether the correlation between gyrification and common genetic variation linked to AOW was present in newborns, we fit a GLM testing for a significant effect of AOW PGS on the mean gyrification index in the left and right hemisphere of the dHCP infants.

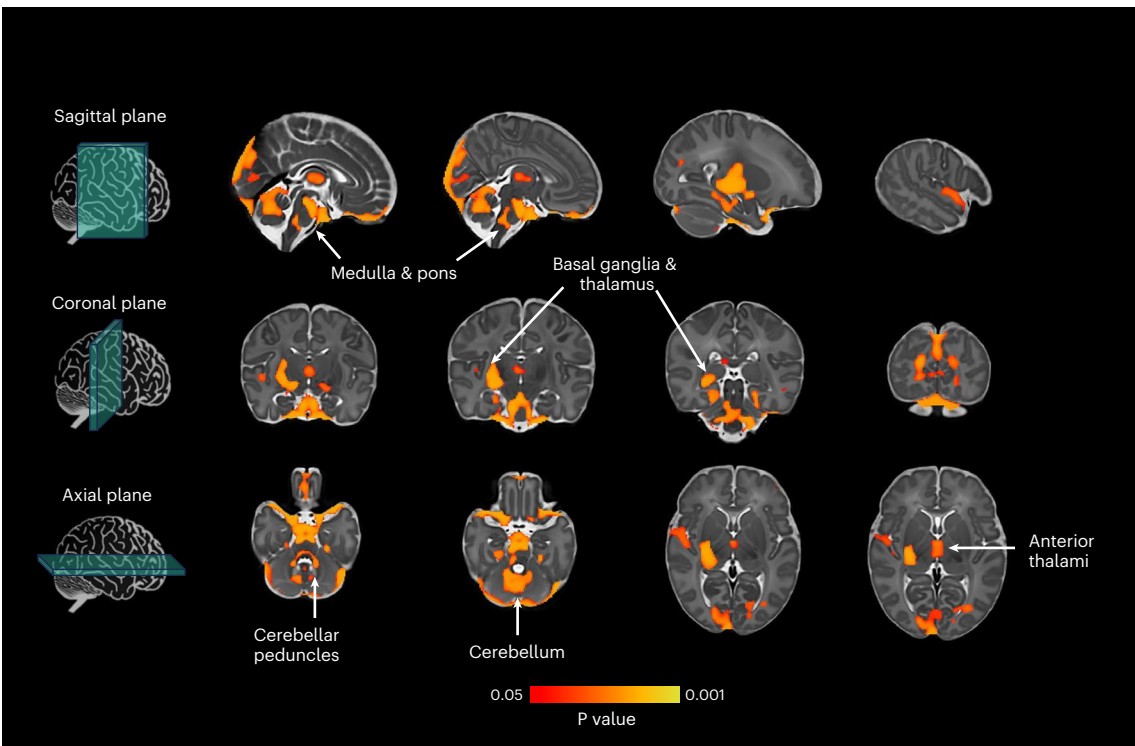

**Fig. 5 | Brain regions with statistically significant positive correlation between tissue volume and age at onset of walking polygenic score in the Developing Human Connectome Project cohort.** Thresholding $t$-statistic image at $t > 0.95$ (two-sided statistical test). Significant voxels were overlaid on the 40-week neonatal brain template in sagittal, coronal and axial planes. White arrows indicate significant brain structures involved in motor control. $N = 264$.

We found a significant positive association between AOW PGS and gyrification index in both hemispheres in newborn brains (left hemisphere $\beta = 83{,}517.30$, CI = 14869.22–152165.39, $P = 0.017$; right hemisphere $\beta = 83{,}839.82$, CI = 18,552.48–149127.15, $P = 0.007$).

Finally, for those infants that had also been assessed using the Bayley-III Scales of Infant and Toddler Development[62] at 18-month-age follow-up ($N = 217$), we explored the relationship between scaled gross motor score and the AOW PGS using a multiple linear regression model. Data distribution was assumed to be normal, but this was not formally tested. We found that higher AOW PGS was significantly associated with lower Bayley's gross motor score, indicating worse/possibly delayed gross motor skills ($t_{(201)} = -2.305$, $\beta = -0.161$, s.e. = 0.070, $P = 0.022$).

## Discussion

The reasons for the high variability in AOW in young children are poorly understood, yet this milestone is used widely as a clinical marker to index overall development, with absence of walking at age 18 months prompting clinical referral to a paediatrician for further assessment and investigation. The present study reveals that AOW is a heritable polygenic trait with significant aetiological links to later health outcomes. Moreover, we identified 11 independent genome-wide significant loci associated with AOW, one of which co-localized with eQTLs and was located in a gene associated with rare disorders that include delayed or absent walking. We discuss four main conclusions from these results.

Past models of gross motor skills, as well as neurodevelopment more generally, have put a primary emphasis on environmental factors such as nutrition[10] and cultural factors[9,63]. Our first conclusion is that our results show that AOW is also associated with common genetic variants operating in the brain. Significantly enriched cell-type tissues were exclusively brain-based tissues; moreover, strongest signals included tissues in the basal ganglia, cortex and cerebellum. In line with these findings, the polygenic score for AOW was associated with neonatal brain volume of the basal ganglia, thalami, medulla, pons and cerebellum. This is consistent with the known role of these brain areas in motor function[11,64]. Also supporting this first conclusion, we found that gene sets involved in AOW are also involved in the generation of neurons. Further, we observed that genes associated with AOW are enriched in the brain between 19 and 24 weeks post conception (Supplementary Fig. 14).

Our second conclusion is that the novel loci that were discovered here involve genes of highly plausible biological relevance to the onset of walking. We identify common variant associations with AOW at a locus overlying RB transcriptional corepressor like protein 2 (*RBL2*, ENSG00000103479, genomic locus 2 in Table 1). *RBL2* is also associated with an autosomal recessive neurodevelopmental disorder (eponym Brunet–Wagner)[65,66]. Homozygous loss of *RBL2* has been observed in five individuals across three families, each with a different allele[65,66]. Affected individuals had infantile hypotonia, severe developmental delay, delayed/absent walking, and were minimally verbal. Seizures were reported in three cases. Three cases had microcephaly (−2.4 s.d. to −4.7 s.d.), while two had normal head circumference (65th and 50th centiles) but cerebral atrophy on magnetic resonance imaging (MRI). Height was normal for two cases, unreported for one and low for two (3rd centile, −3.4 s.d.). In Balb/c mice, homozygous loss of *Rbl2* is embryonic lethal with a disorganized neural tube and neuronal loss[64] (see also Supplementary Note B).

The third conclusion from our results is that AOW is partly influenced by the same genetic variants that influence individual variability of other complex traits measured at later ages. We found that common genetic variation associated with AOW is partly overlapping with common genetic variation associated with cognitive performance and years in education, likelihood of ADHD and cortical folding index. We note that the direction of these associations was consistent in the three largest individual cohorts (MoBa, NTR and Lifelines) (the fourth cohort, NSHD was not well-powered for genetic correlation estimates) as well as the meta-analysed results, indicating robust findings. Interestingly, MiXeR analyses showed that a large proportion of variants explaining the heritability in AOW were shared with educational attainment and

cognitive performance, with more than half of these variants having concordant effects on the two phenotypes (which explains the overall positive genetic correlations obtained with the LDSC method shown in Fig. 4a). Thus, results indicated that genetic predispositions to later onset of walking also contribute to high cognitive performance and more educational attainment. It is interesting to note that nearly half of the overlapping SNPs between AOW and cognitive performance and academic achievement have discordant effects.

The negative genetic correlation between AOW and ADHD might be surprising when considering that, at the phenotypic level, delayed walking, rather than earlier walking, is associated with increased likelihood of developmental disorders[4]. However, the ability to walk requires practice and movement[63], and infants with higher activity levels or shorter attention spans may, on average, move about more, thus gaining more practice in movement, muscle strengthening and training, ultimately resulting in earlier walking onset. Thus, attention and activity levels may influence motor system training in young children, and this may relate to what we are observing at the level of common genetic variation. In support of the hypothesis that shorter attention span and higher activity levels would be associated with earlier walking, a recent study of over 25,000 children from MoBa found that the ADHD polygenic score was associated with earlier walking[18]. Further, the ADHD polygenic score was associated with better gross motor skills, such as walking, climbing stairs and jumping, in 7,498 18-month-old children from the Avon Longitudinal Study of Parents and Children (ALSPAC)[67]. At the same time, it should be noted that in our study, the negative genetic correlation between AOW and ADHD, while significant, is still relatively modest in magnitude ($r_g = -0.180$).

Research on the timing of milestones in prenatal brain development across humans, primates and other mammals shows that longer duration (more prolonged development) is associated with larger brain volumes, and in particular, enlargement of later developing brain structures[68]. In line with this, within humans we found that the polygenic score predisposing to later onset of walking is associated with larger volumes of neonatal brain areas involved in the motor domain (Fig. 5). In addition, we found that gene sets associated with AOW are involved in neurogenesis, and that expression of genes associated with AOW is enriched in the brain between 19 and 24 weeks post conception (Supplementary Fig. 14). Last, we found that later AOW is genetically correlated with increased cortical folding in adolescence and adulthood in areas involved in the somatosensory processing of movement (ROIs 1 and 5 m in Glasser parcellation[56], located in the primary somatosensory cortex), including higher-order somatosensory integration of the lower limb representation (ROI 24dd in the cingulate motor area) and motor planning concerning the whole body (ROI 6r in the premotor cortex). Taken together, these findings may suggest that for children with genetically influenced protracted subcortical neurogenesis in the prenatal period, cortical regions involved in more complex motor behaviours may take longer to specialize[69]. This results in a later onset of walking. Since advantages and costs to early walking might vary on the basis of the individual's environmental conditions, wide individual differences in the duration of the sensitive period to learn to walk might be the result of the ability of human beings to adapt to their local environment[70].

Current public health policy employs late walking (>18 months) as a red flag for developmental delay which typically triggers referral for clinical assessment aimed to identify the reason for a departure from the normal range of achievement of this milestone[3]. A better understanding of the entire variation of AOW and of its shared biology with later medically relevant phenotypes could help in more precise intervention planning. Future research should test whether adding AOW PGS to clinical variables and/or rare variant information could improve prediction models that could be applied clinically. Historical data suggest that the majority of late walkers do not have a medically recognized developmental disorder[6]. In light of our findings, future research should explore whether early walking may also be a useful red flag that may offer early information about likelihood of ADHD or learning difficulties.

Our final conclusion is that the genetic signal identified through our AOW GWAS captures genetic effects that directly influence the phenotype[48]. This was tested by the within-family polygenic score analyses on fraternal twin siblings in the NTR cohort. We found that the variance explained by the between-pair PGS was not significantly greater than that explained by within-pair association. If the variance explained by between-pair PGS had been much larger than the within-pair PGS, it would have indicated that some of the AOW signal was coming from genetic effects that play a role on the phenotype in an indirect way, via mechanisms such as gene–environment correlation, assortative mating and stochastic effects[48]. Our results offer evidence that the polygenic score is picking up on direct genetic effects.

In our study design, we took a comprehensive approach to the phenotype and samples. Relevant samples were searched for using multiple database resources, research council websites and bibliographies. Samples were only included if they had a highly similar phenotype (AOW in months) and a sample size greater than 1,000 to ensure reliable effect sizes in individual samples. Nevertheless, the potential attrition and participation biases present in population cohorts should be considered in relation to our findings[71,72]. Although there is evidence that AOW can be reliably recalled by parents retrospectively by the child's second[1] and third[18] birthday, we acknowledge that it was not possible to measure the reliability of this phenotype as recorded in the Lifelines cohort, where it was collected between the children's 3 and 18 years of age. It is possible that the Lifelines measure included the largest measurement error of the four cohorts, in light of the later age at which parents recalled the AOW in their children (although AOW difference between cohorts was not correlated with the interval between AOW and parent report). Systematic regional/national differences in body size and cultural factors might explain these differences. However, by conducting the GWAS in each individual cohort and then meta-analysing the effects, our approach ensures capturing SNP effects on the trait variance that are not confounded by differences between cohorts. An important limitation of this study is that our meta-analysis only included Western European cohorts, as at the time of conducting the study, information on AOW was not available in other sufficiently large genotyped cohorts to have the statistical power for a GWAS. Extending this investigation to a more diverse population is a vital next step. Future work could also test the degree to which genetic correlations with AOW vary locally across the genome, and furthermore, how they vary when conditioned on third variables to delineate genetic associations with AOW within specific genomic locations[73].

In summary, we demonstrate that the high variability in age at onset of walking is partly due to common genetic variation, with approximately a quarter of the variability explained by common genetic variants. The genetic variants identified were plausible contributors to individual variability in motor behaviour, as they were previously associated with disorders that disrupt the development of walking. AOW was shown to be an important milestone that links genetically to a range of later health, educational and behavioural outcomes.

## Methods

### Inclusion and ethics statement

This study complies with all relevant ethics regulations. The research and the related secondary data analysis were approved by the Departmental Ethics Committee of the Psychological Science Department of Birkbeck, University of London, on 27 October 2020 (reference number 2021007). Each cohort received ethics approval by the local ethics review committee. The current research was not conducted in resource-poor settings. Researchers responsible for the cohort data management in each of the three European countries (Norway, the Netherlands, United Kingdom) were involved in the research process

and consulted regarding authorship and relevant citations. No biological materials were shared for the purpose of this study. This study did not involve animal data.

MoBa and the related data collection was authorized by a licence from the Norwegian Data Protection Agency and an approval from the Regional Committees for Medical and Health Research Ethics (REK). MoBa is regulated by the Norwegian Health Registry Act. Informed consent was provided by all participating parents at recruitment. The current study was approved by REK (2016/1702). An external collaborator form was signed for accessing this dataset, and MoBa genotype and phenotype data were shared within a secure server in Norway, in accordance with Collaboration and Data Processor Agreement 20220801 between the Norwegian Institute of Public Health and Birkbeck College, University of London.

Informed consent for NTR was obtained from parents or guardians. The study was approved by the Central Ethics Committee on Research Involving Human Subjects of the VU University Medical Centre, Amsterdam, an Institutional Review Board certified by the US Office of Human Research Protections (IRB number IRB00002991 under Federal-wide Assurance FWA00017598; IRB/institute codes, NTR 03-180). No application for accessing this dataset was needed because for this study, only summary statistics and no personal data were shared between collaborating authors.

Participants in Lifelines gave written consent before physical examination. The study was conducted according to the principles of the Declaration of Helsinki and in accordance with the University Medical Center Groningen (UMCG) research code, and was approved by the Medical Ethics Committee of UMCG (document number METC UMCG METc 2007/152). Data were accessed in accordance with Material Transfer and/or Data Access Agreement OV19_0511 between Medische Biobank Noord Nederland B.V. for Lifelines and Birkbeck College, University of London.

For NSHD, the collection of blood samples and DNA information from the participants was approved by ethics approval reference MREC no. 98/2/121. No application for accessing this dataset was needed because for this study, only summary statistics and no personal data were shared between collaborating authors.

The Developing Human Connectome Project was approved by the UK Health Research Authority (Research Ethics Committee reference number: 14/LO/1169), and written parental consent was obtained in every case for imaging and open data release of the anonymized data. No application for accessing this dataset was needed because no personal data were shared between collaborating authors.

### Samples
The meta-analysis was conducted using data from four birth cohort samples of European ancestry. Full details of the samples are provided in Supplementary Note A.

Analyses were preregistered on the Open Science Framework on 24 February 2023 (https://doi.org/10.17605/OSF.IO/M2QV3).

**The Norwegian Mother, Father and Child Cohort Study.** MoBa is a population-based pregnancy cohort study conducted by the Norwegian Institute of Public Health[20,21]. Participants were recruited from all over Norway from 1999–2008. The women consented to participation in 41% of the pregnancies. Blood samples were obtained from both parents during pregnancy and from mothers and children (umbilical cord) at birth[74]. The cohort includes ~114,500 children, 95,200 mothers and 75,200 fathers. The current study is based on version 12 of the quality-assured data files released for research in January 2019. Phenotype information used in this study (year of birth and sex of the participants) was obtained from the Medical Birth Registry (MBRN), a national health registry containing information about all births in Norway.

After post-imputation quality control, the MoBa dataset included 207,569 individuals, of whom 76,577 were children[75]. The final sample size of children from MoBa with European genetic ancestry and good-quality genotype and phenotype information included in the GWAS was 58,302 (28,456 females, 29,846 males).

**Netherlands Twin Register.** The NTR consists of twins, multiples and their family members. NTR twins and multiples were recruited into the register as newborns up to a few months after birth starting in 1987 (ref. 76). There were no exclusion criteria. Genotyping was performed on 7,392 individuals for whom there was parent-report data in infancy[77]. For NTR, 6,251 children (3,399 females, 2,852 males) with good-quality genotype and available phenotype data were included in the GWAS.

**Lifelines.** Lifelines is a multigenerational prospective population-based birth cohort study examining the health and health-related behaviours of 167,729 persons living in the North of the Netherlands. It employs a broad range of investigative procedures in assessing the biomedical, socio-demographic, behavioural, physical and psychological factors that contribute to the health and disease of the general population, with a special focus on multimorbidity and complex genetics[23]. Individuals aged 25 to 50 were recruited from the Northern region of the Netherlands between 2006 and 2013 and, during their first study visit, were asked for consent for the study team to approach family members with an invitation to participate. This included any children (≥6 months) of cohort members. Questionnaires about children were answered by parents on the basis of retrospective recollection. The final sample size of Lifelines children with good-quality phenotype and genotype data included in the GWAS was 3,415 (1,768 females, 1,647 males).

**MRC National Study for Health and Development.** NSHD is a population-based prospective birth cohort study whose participants were infants from single births born in England, Scotland and Wales during 1 week in March 1946 ($N = 5,362$) to women with husbands[24]. The dataset included 2,939 genotyped individuals whose DNA was collected at age 53 (ref. 78). The sample was roughly representative of the national population of the same age at the time according to a comparison with census data. The final NSHD GWAS sample size including children with available genotype and phenotype was 2,592 (1,295 females, 1,297 males).

### Phenotype coding
In all samples, individuals whose AOW was less than 6 months or greater than 36 months were excluded as outside the normative range[3]. MoBa, NSHD and NTR all recorded AOW in months as an integer variable. In the Lifelines sample, age at onset of walking was measured as an ordinal scale, using bins of months of age at onset of walking. These were recorded using the midpoint for each age bin. The upper and lower bins ('10 months or younger' and '24 months or older', respectively), were winsorized, recoding them to 10 and 24 months, respectively. The phenotype descriptives for each cohort are reported in the Supplementary Table 1. Normality and spread of the phenotype data distribution was formally tested. All four cohorts met the assumptions of normality in terms of symmetry of the distribution (skewness = 0.43–0.91, see Supplementary Table 1). NSHD (kurtosis = 3.88), MoBa (kurtosis = 3.26) and, to a lesser degree, Lifelines (kurtosis = 1.33) showed a peaked distribution, different from NTR (kurtosis = −0.12). Histograms for the phenotype data distributions are reported in the Supplementary Notes (Supplementary Figs. 1, 3, 5 and 7).

### Genotyping, imputation and quality control
Pre- and post-imputation quality control (QC) and imputation procedures were conducted for each cohort following individual study protocols and according to a common standard operating procedure (https://osf.io/jyk6d/), which was based on the Rapid Imputation for COnsortias PipeLIne (RICOPILI) pipeline[79]. In all the individual cohorts, samples were excluded from the GWAS if they presented

excess autosomal heterozygosity, mismatch between self-reported and genetic sex, XXY genotype and other aneuploidies, and individual genotyping rate <90% in line with established GWAS analysis pipelines[79,80]. Duplicate samples and samples whose genetically determined ancestry did not overlay with the European-ancestry cluster based on a reference panel were also excluded to minimize confounding due to population stratification. Autosomal SNPs were excluded from the GWAS if they had MAFs < 0.5%, Hardy–Weinberg equilibrium exact test at $P < 1 \times 10^{-6}$ and call-rate <98%. Full details of the pre- and post-imputation QC are provided in Supplementary Note A and Table 2.

### Genome-wide association analyses
GCTA[81] fastGWA[82] was used for association analyses in MoBa, Lifelines and NTR. PLINK[83] 1.9 was used for association analyses in NSHD, where all related individuals (PI-HAT > 0.2) were excluded from the analysis and the sample size was too small to use fastGWA.

Association analyses of the AOW, as a continuous variable, were carried out using a mixed linear model. Each primary GWAS included the first 10 ancestry principal components as continuous covariates, and sex and genotyping batch as discrete covariates. MoBa included year of birth, and NTR and Lifelines included age at data collection as continuous covariates. NTR included two dummy variables for the genotyping platform as covariates. In MoBa, Lifelines and NTR, where fastGWA was used, a sparse (0.05 cut-off) genetic relatedness matrix was included in the model to account for relatedness in the sample.

GWAS analyses were performed for each of the samples using the whole dataset and also with the samples stratified by sex.

### GWAS meta-analysis
Summary statistics QC was performed using the GWASinspector[84] R package on each of the cohorts' summary statistics separately. Variants were excluded if they (1) presented invalid or missing values in the chromosome, position, effect and other allele, beta, standard error columns, and duplicated alleles; (2) were monomorphic (with allele frequency of 0 or 1 and variants with identical alleles), allosomal or mitochondrial; or (3) had imputation quality score <0.8. Results of the summary statistics QC are provided in Supplementary Note A and Table 3.

Summary statistics for the four samples were meta-analysed with a standard-error-weighted meta-analysis in METAL[28] on SNPs with MAF > 1%. SNPs were matched between cohorts using rsIDs, which had been assigned according to their chromosome, base-pair positions and alleles on the basis of the 1000 Genomes[85] reference panel in GWA-Sinspector. Meta-analyses were performed separately for the whole sample and for sex-stratified samples. Finally, only SNPs for which the minimum sample size was 10,000 (which was obtained if the SNP was available for the MoBa sample, all three other cohorts or if it overlapped in all four cohorts) were retained for further analyses (6,902,401 variants). The $I^2$ heterogeneity metric per SNP was calculated in METAL. $M$ multiSNP heterogeneity statistics, indicating whether individual studies were systematically more influential or weaker than average based on their effects, was calculated using the getmstatistic R package for the independent lead SNPs (pairwise LD $r^2 < 0.1$, $P < 5 \times 10^{-8}$, $N$ SNPs = 16)[31].

### Fine mapping and functional annotation
To identify significant independent SNPs associated with AOW at each locus at a $P$-value threshold of $P < 5 \times 10^{-8}$ (ref. [86]), we conducted conditional and joint association analyses (COJO)[29] in GCTA[81]. This analysis conditions on the lead SNP at a locus and tests for further independent significant SNPs within the same chromosome using a stepwise selection procedure. The MoBa genotype data were used to estimate linkage disequilibrium (LD), in line with the COJO guidelines.

Fine mapping, functional annotation and gene-based analyses were carried out in FUMA[33] (v.1.5.2) and MAGMA[37] (v.1.08), indicating the list of independent lead SNPs from the COJO analysis. We defined

significant SNPs to be independent if they had pairwise LD $r^2 < 0.6$. Lead SNPs were defined as having pairwise LD $r^2 < 0.1$ (ref. [87]). Loci were merged if LD blocks distance was <250 kb.

For gene-mapping in FUMA, SNPs were mapped to genes at a maximum distance of 1 Mb[33] on the basis of position, eQTL for selected relevant tissues such as the brain, lung, muscles, heart and adipose tissue, and chromatin interaction in the brain (see Supplementary Table 6). Annotation of genes was performed using ANNOVAR within FUMA (date of download 17 July 2017).

A subset of genes prioritized on the basis of mapping using only significant SNP–gene pairs at an FDR corrected $P < 0.05$ were tested for differential expression in 54 Genotype-Tissue Expression (GTEx) (v.8)[35] and 11 BrainSpan[34] tissues, and gene-set enrichment using GENE2FUNC in FUMA. The gene-set analysis in FUMA used one-sided hypergeometric tests to test whether the prioritized genes were over-represented in predefined gene sets obtained from the Molecular Signatures Database[88,89] (MSigDB) v.7.0, WikiPathways[90] (v.20191010) and GWAS Catalog[91] (v.e0_r2022-11-29) databases, after excluding the MHC region and applying Bonferroni correction for multiple testing.

For MAGMA analyses, the MHC region was excluded and SNPs within 1 kb from a gene were assigned to each gene[87]. The MAGMA gene-based test identified genes associated with AOW from all 18,766 mapped genes using a Bonferroni correction to define statistical significance (Supplementary Table 9). The MAGMA gene-property analysis used 53 GTEx (v.8)[35] and 11 BrainSpan[34] RNA-seq datasets to test tissue specificity of genes associated with AOW, based on association one-tailed $P$ values of all 18,766 genes mapped in FUMA.

### Co-localization
We used coloc SuSiE[43] to identify co-localization of GWAS and eQTL signals, using an LD reference panel of 1,444,196 HapMap3 SNPs with LD calculated in European-ancestry individuals from the UK Biobank[92,93]. Pairs of variants further than 3 cM apart were assumed to have 0 correlation. We used coloc SuSiE's default priors (for more information on how these priors were estimated, see ref. [94]). The eQTL data used in the co-localization analyses were from 261 post-mortem bulk RNA-seq samples of human cerebellum[42]. We replicated the co-localization signal observed in *RBL2* (Fig. 2b) in the human cortex using eQTL data from 1,433 post-mortem bulk RNA-seq samples[42] (Supplementary Fig. 15). To validate in an independent dataset whether genotype was indeed associated with *RBL2* expression, we used bulk RNA-seq data of prefrontal cortex and individual-level genotypes from BrainVar[44] (periods 4–6; Fig. 2d) (as no publicly available cerebellum RNA-seq with genotype on the same individual exists, to our knowledge). We used a two-sided Wilcoxon rank test to test for differences in *RBL2* expression in the human cortex by genotype for GWAS and eQTL significant SNPs at MAF ≈ 50%. Missense variants in the chromosome 16 locus were annotated using the Variant Effect Predictor (VEP)[45]. The protein structure for *RBL2* was predicted using AlphaFold[95]. Annotation of p.Tyr210Cys on *RBL2* was done using the Genomics 2 Proteins Portal[96].

### LD score regression
LD score regression (LDSC[25]) was used to calculate $h^2_{SNP}$ and bivariate genetic correlations[30], using the 1000 Genomes Phase 3 (ref. [85]) European-ancestry LD scores reference panel. Bivariate genetic correlations were calculated between AOW and multiple infant, psychiatric, neurodevelopmental and global cortical phenotypes, specifically: birth weight[97], childhood body-mass index (cBMI)[49], adult BMI[50], autism[98], ADHD[51], educational attainment (EA)[52], cognitive performance[53], schizophrenia[99], general loading for psychiatric disorders (cross-disorders)[100], major depression[101], bipolar disorder[57] and 13 cortical phenotypes[54] (see Fig. 4a). Genetic correlation was also calculated between the AOW in each of the cohorts.

In addition, LDSC was used to calculate $h^2_{SNP}$ for the female and male meta-GWAS and genetic correlation between the sex-stratified

analyses. Statistical significance was evaluated on the basis of 95% confidence intervals as preregistered. As post-hoc analyses, which were not preregistered, we also used LDSC to test the genetic correlation between AOW and four other motor phenotypes: self-reported walking pace[55], clinically ascertained muscle weakness in the pincer grip in elderly people[102], motor coordination in childhood[103] and Parkinson's Disease[104]. Bonferroni-adjusted *P* values correcting for 28 multiple testings are reported in Supplementary Table 14.

To further investigate the significant genetic correlation between AOW and cortical folding index (FI), we ran non-preregistered genetic correlation analyses using 26 regional FI summary statistics from ref. 54. The 26 ROIs were defined following the Glasser parcellation and identified on the basis of their functional specialization as early somatosensory/motor areas according to ref. 56. Given that regional FI could not be assumed to be completely unrelated, we applied FDR correction for 26 simultaneous tests.

Stratified LDSC[40] was conducted to obtain estimates of heritability partitioned by functional annotation and cell-type. HapMap3 (ref. 105) SNPs (excluding the HLA region) from the meta-GWAS summary statistics weighted by LD score obtained from a European 1000 Genomes[85] reference panel were used in the regression, as recommended by ref. 40. To estimate the proportion of genome-wide $h^2_{SNP}$ attributable to functional categories, we ran the stratified LDSC 'full baseline model' (described in ref. 40) that evaluates whether heritability in a functional category is greater than heritability outside the category. This was tested for 96 functional categories provided by the stratified LDSC developers, including coding, untranslated regions, promoter and intron annotations from UCSC[106], genomic annotations for all cell types and fetal cell types only from ENCODE[107] and the Roadmap Epigenomics Consortium[108], region conserved in mammals from ref. 109 and FANTOM5 enhancers from ref. 110. The *P* value for enrichment was adjusted for multiple testing using the Bonferroni method, as in similar previous research[111].

To calculate whether heritability was enriched in specific cell types, we applied stratified LDSC to 53 sets of specifically expressed genes[41] using multitissue gene expression data from the GTEx[35] project. Bonferroni correction was applied to correct for multiple testing.

### Genomic Structural Equation Modelling (SEM)

A non-preregistered Genomic SEM[58] analysis was conducted to test whether the association of the genetic components of AOW with ADHD remained significant after conditioning for educational attainment. To this aim, we performed a genetic multivariable regression using the same ADHD[51] and EA[52] summary statistics that were entered in the LDSC analysis. For ADHD, the sample size was defined as effective sample $N_{eff} = 4 \, v \times (1-v) \times (N_{cases} + N_{controls})$ where $v$ was the sample prevalence set as 50%, as indicated by the Genomic SEM developers (https://github.com/GenomicSEM/GenomicSEM/wiki/2.-Important-resources-and-key-information). The summary statistics were munged using HapMap3 SNPs. Both standardized and unstandardized results are reported in Supplementary Fig. 16.

### MiXeR

Univariate causal mixture models were applied using MiXeR[59] to obtain estimates of polygenicity, defined as the proportion of variants that contribute to 90% of the $h^2_{SNP}$[112]. We fitted bivariate models in MiXeR to estimate the genetic overlap that was due to both concordant and discordant SNP effects between AOW and six other phenotypes that had a Bonferroni-significant genetic correlation with AOW (calculated using LDSC). For each pair of traits, the models were evaluated using differential BIC and AIC values between the 'best' bivariate model estimating the optimal amount of polygenic overlap between the two traits (grey areas in Fig. 4b) and two simpler models, namely, the 'minimum' and the 'maximum' overlap models. The 'minimum' overlap models used only the minimum number of SNPs to explain the

genetic overlap from the LDSC genetic correlation estimate, while the 'maximum' overlap models assumed that all the variants associated with the least polygenic of the two traits overlapped with the other trait. Positive differential BIC and AIC values indicated the 'best' MiXeR bivariate model outperforming the two simpler models. When the summary statistics for the second phenotype in these bivariate analyses came from the case-control GWAS, the $N_{eff}$ was calculated as $4/(1/N_{cases} + 1/N_{controls})$. The MHC region (6:26,000,000–34,000,000) was excluded from MiXeR analyses due to its complex LD structure, in line with the programme recommendations. MiXeR v.1.3 was used for these analyses, and the data were prepared using scripts developed by the programme's authors (https://github.com/precimed/python_convert).

We considered the bivariate MiXeR model to be supported when the differential AIC value comparing the 'best' vs 'minimal' model was positive. This criterion ensures that there is support for the model of the polygenic overlap that includes the added free parameters of this model.

### Polygenic score analysis

Polygenic scores were calculated using PRS-cs[113,114]; a leave-one-out design was employed whereby additional GWAS meta-analyses were conducted, leaving out one of each of the smaller samples (NSHD, NTR and Lifelines) in turn to be used as a target dataset and meta-analysing the remaining samples as a training dataset for estimation of SNP weights. The MoBa sample comprises most of the overall sample size and thus could not be used as a target dataset, so a within-MoBa cross-validation was employed. The MoBa dataset was split randomly into five samples of roughly equal size by removing one-fifth of the data in turn (with no overlap in these fifths) from the whole dataset to create five new samples, each comprising four-fifths of the data. GWASs were then conducted on each of these five new samples and the summary statistics of the meta-analysis of four samples used for estimation of PGS SNP weights applied to the left-out fifth of the data. This was performed five times, using each of the fifths as target data in turn.

For all leave-one-out PGS analyses, including the within-MoBa design, we derived weights for each chromosome using the 1000 Genomes phase 3 European panel[85] as a reference for LD, and the following PRS-cs parameters: parameter *a* and *b* in the gamma-gamma prior = 1 and 0.5, respectively, global shrinkage parameter phi = 0.01, 1,000 MCMC iterations, 500 burn-ins and 5 as a thinning factor of the Markov chain. PLINK (2.0)[115] was used to compute the PGS in the target sample. The proportion of variance explained by the PGS, scaled so that mean = 0 and s.d. = 1, was quantified in the NTR cohort by the squared beta-coefficient from a linear regression model between the scaled phenotype and the PGS, including 10 ancestry principal components (PCs), age, sex and genotyping platform in the model, and quantified in all other cohorts with adjusted $R^2$ of the linear regression between the scaled phenotype regressed on 10 PCs and the genotype batch and the PGS.

### Within- and between-family polygenic score analysis

Within- and between-family analyses were performed using the NTR cohort dataset. The method is described in ref. 48 and scripts from ref. 116 were used (https://github.com/PerlineDemange/GeneticNurtureNonCog/).

A PGS was generated from a meta-analysis of the MoBa, Lifelines and NSHD GWAS (calculated as above), and the predictive power of this PGS was quantified in the whole NTR sample using the above method. We used a random intercept mixed-effects linear model in R using the dizygotic twins-only subsample of NTR (*N* = 2,508 individuals in 1,254 twin pairs), after ensuring that a mixed-effects model was justified by calculating a bootstrapped intraclass correlation (ICC = 0.656) as indicated in ref. 48. PGS entered into the model were first scaled to mean = 0 and s.d. = 1. Within-family PGS effects were calculated by subtracting the family mean PGS from each individual PGS. Between-family effects

were modelled using the mean PGS for each family. The linear model included age, sex, the first 10 PCs and a genotyping platform dummy variable as covariates. The within- and between-family standardized regression coefficients were compared using a $\chi^2$ test.

**Polygenic score in the Developing Human Connectome Project**

**Genetic data.** Infant saliva DNA was genotyped for SNPs genome-wide on the Illumina Infinium Omni5-4 array and standard quality control was performed. The dataset was imputed to the Haplotype Reference Consortium reference panel[117] on the Michigan Imputation Server. The imputed data were used to compute an AOW PGS for each of the 264 unrelated European infants using summary statistics from the AOW meta-GWAS and the PRS-cs software[113], as previously described.

**Acquisition, processing and surface generation of imaging data.** T2-weighted MRI data were acquired at term-equivalent age (median postmenstrual age = 41.9 weeks) as part of the dHCP[60] in 264 term-born infants (137 male, 127 female) with available genotype data. The volumes were run through the neonatal-specific processing pipeline developed for the dHCP study, including bias field correction, brain extraction and image segmentation[118–120]. Segmentations were used to generate cortical, white matter and pial surfaces, and each subject was visually inspected to ensure accuracy before the local gyrification index was calculated at each vertex on the basis of the ratio of the pial and white matter surface areas[121,122].

**Image registration.** T2 images were registered to the 40-week dHCP neonatal atlas (https://brain-development.org/brain-atlases/atlases-from-the-dhcp-project/)[123] via an age-matched intermediate using Symmetric Diffeomorphic Image Registration, implemented using Advanced Neuroimaging Tools (ANTs)[124,125], as a measure of individual variation in brain volume; the log-Jacobian determinant images were calculated by applying ANTs algorithms to the nonlinear transformation deformation tensor fields. Log-Jacobian maps were then smoothed using a 3-mm full-width half-maximum Gaussian filter and downsampled to 1 mm isotropic resolution (to increase computational efficiency). A 4D volume was created by merging the 1-mm log-Jacobian maps across all participants ($N = 264$), then subsequently used as the input to the randomize algorithm (described below).

**Tensor-based morphometry of imaging data.** Permutation testing using the randomize function, part of the FMRIB Software Library (FSL)[126,127], was used with a general linear model, including gestational age, postmenstrual age at scan, sex, weight $z$-score and 10 ancestral PCs as covariates. Threshold-free cluster enhancement and family-wise error (FWE) rate were applied to correct for multiple comparisons between voxels. Significant areas were identified with permutation testing using 5,000 random permutations (two-sided test). In Fig. 5, we show results at a significance level of $P < 0.05$ in the FWE-corrected contrast.

**Bayley's gross motor analysis.** For the European term-born infants in the dHCP cohort who were assessed using the Bayley-III Scales of Infant and Toddler Development at an 18-month follow-up ($N = 217$), we investigated the association between the scaled gross motor score and the PGS for AOW using a multiple linear regression model, implemented using the lm function in R (https://www.r-project.org/). The model included sex, gestational age at birth, birth weight $z$-score, home environment score (as a proxy for socioeconomic status) and 10 ancestral PCs as covariates to account for potential confounding. All continuous variables were standardized before analysis. Data distribution was assumed to be normal, but this assumption was not formally tested.

**Reporting summary**

Further information on research design is available in the Nature Portfolio Reporting Summary linked to this article.

## Data availability

The summary statistics of the genome-wide association study of age at onset of walking are available on figshare (https://doi.org/10.6084/m9.figshare.28071566)[128]. Data from the Norwegian Mother, Father and Child Cohort (MoBa) Study and the Medical Birth Registry of Norway used in this study are managed by the National Health Register Holders in Norway (Norwegian Institute of Public Health) and can be made available to researchers, with approval from the Regional Committees for Medical and Health Research Ethics (REC), compliance with the EU General Data Protection Regulation (GDPR) and approval from the data owners. The consent given by the participants is not open to storage of data on an individual level in repositories or journals. Researchers who want access to datasets for replication should apply through https://helsedata.no/. Access to datasets requires approval from The Regional Committee for Medical and Health Research Ethics in Norway and an agreement with MoBa. Data from the Netherlands Twin Register (NTR) are available upon request by researchers. Information is available at https://tweelingenregister.vu.nl/information_for_researchers/working-with-ntr-data. Lifelines data may be obtained from a third party and are not publicly available. Researchers can apply to use the Lifelines data used in this study. More information about how to request Lifelines data and the conditions of use can be found on their website at https://www.lifelines-biobank.com/researchers/working-with-us. National Study for Health and Development (NSHD) data used in this publication are available to bona fide researchers upon request to the NSHD Data Sharing Committee via a standard application procedure. Further details can be found at http://www.nshd.mrc.ac.uk/data. https://doi.org/10.5522/NSHD/Q101. eQTL results for the ROSMAP, Mayo TCX, Mayo CER and cortical meta-analysis from ref. 42 are available through the AMP-AD Knowledge Portal: https://www.synapse.org/Synapse:syn2580853/wiki/409840. The accession number for the raw RNA-seq and WGS data from BrainVar, along with processed files, is PsychENCODE Knowledge Portal: syn21557948 on Synapse.org (https://www.synapse.org/#!Synapse:syn4921369). Developing Human Connectome project data are open access and data are available for download via https://nda.nih.gov/edit_collection.html?id=3955.

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

## Acknowledgements

This work was funded by the Simons Foundation for Autism Research Initiative (SFARI, 724306) to A.R.; the Economic and Social Research Council (ES/P000592/1) to A. Hollowell; the South-Eastern Norway Regional Health Authority (2019097 and 2022083 to L.J.H.; 2021045 to E.C.C.; 2020022, 2024001 and 2019097 to A. Havdahl); the Research Council of Norway (274611 to E.C.C.; 274611 and 336085 to A. Havdahl; 324499, 324252, 223273 to O.A.A.); the UK Medical Research Council (MRC, MC_UU_00019/1 to A.W.; MR/V036874/1 and MR/N026063/1 to T.A.; MRC PG MR/T003057/1 to M.H.J.; MR/S037055/1 to F.D.); a UK National Institute for Health Research (NIHR) fund to H.C.; SFARI, Wellcome Trust (214322\Z\18\Z), Horizon-Europe R2D2-MH (101057385), and UKRI (10063472) to V.W.; KG Jebsen Stiftelsen, Nordforsk (164218), EU H2020 RIA grant 964874 REALMENT to O.A.A.; HORIZON-HLTH-2021 R2D2-MH (101057385) to B.S.P.; ERC-2017-COG (771057) WELL-BEING and NWO Vici scheme (VI.C.211.054 504 'The Power of Wellbeing') to M.B.; a KNAW Academy Professor Award (PAH/6635) to D.I.B.; the European Research Council (FP/20072013 to A.D.E.); the National Institute of Mental Health (NIMH, R01MH129751 and U01MH122681) to S.J.S.; HDR UK QQ2 Molecules to Health Records Driver Programme to S.J.S; and European Union's Horizon Europe Research and Innovation programme Marie Skłodowska-Curie grant agreement European Social Science Genetics Network (101073237) to A. Havdahl. The funders had no role in study design, data collection and analysis, decision to publish or preparation of the manuscript. The views expressed are those of the authors and not necessarily those of the MRC, the NIHR or the Department of Health and Social Care. The Norwegian Mother, Father and Child Cohort Study is supported by the Norwegian Ministry of Health and Care Services, and the Ministry of Education and Research. We thank all the participating families in Norway who take part in this on-going cohort study; the Norwegian Institute of Public Health (NIPH) for generating high-quality genomic data. This research is part of the HARVEST collaboration supported by the RCN (grant no. 229624). For providing genotype data, we also thank the NORMENT Centre (funded by the RCN (223273), the South-Eastern Norway Regional Health Authority (SENRHA) and Stiftelsen Kristian Gerhard Jebsen), in collaboration with deCODE Genetics, and the Center for Diabetes Research at the University of Bergen (funded by the ERC AdG project SELECTionPREDISPOSED, Stiftelsen Kristian Gerhard Jebsen, Trond Mohn Foundation, the RCN, the Novo Nordisk Foundation, the University of Bergen, and the Western Norway Regional Health Authority). This work was performed on the TSD (Tjeneste for Sensitive Data) facilities, owned by the University of Oslo, operated and developed by the TSD service group at the University of Oslo, IT-Department (USIT) (tsd-drift@usit.uio.no). The computations were performed on resources provided by Sigma2—the National Infrastructure for High Performance Computing and Data Storage in Norway. The Lifelines initiative has been made possible by subsidy from the Dutch Ministry of Health, Welfare and Sport, the Dutch Ministry of Economic Affairs, the University Medical Center Groningen (UMCG), Groningen University and the Provinces in the North of the Netherlands (Drenthe, Friesland, Groningen). We acknowledge the services of the Lifelines Cohort Study, the contributing research centres delivering data to Lifelines, and all the study participants. The MRC National Survey of Health and Development is funded by the UK Medical Research Council (MC_UU_00019/1). We also thank the study participants for their continuing participation in the National Study of Health and Development, and also the study members from the MRC NSHD for their lifelong commitment to the study. The Netherlands Twin

Register acknowledges funding from the Netherlands Organization for Scientific research (NWO), including NWO-Grants NWO/SPI 56-464-14192 and 480-15-001/674: Netherlands Twin Registry Repository and the Biobanking and Biomolecular Resources Research Infrastructure (BBMRI–NL, 184.021.007 and 184.033.111); Amsterdam Public Health (APH) and Neuroscience Campus Amsterdam (NCA); the European Community 7th Framework Program (FP7/2007-2013): ENGAGE (HEALTH-F4-2007-201413) and ACTION (9602768) and European Research Council (ERC-230374). We also acknowledge The Rutgers University Cell and DNA Repository cooperative agreement (NIMH U24 MH068457-06); the Collaborative Study of the Genetics of DZ twinning (NIH R01D0042157-01A1); the Developmental Study of Attention Problems in Young Twins (NIMH, R01 MH58799-03); Major depression: stage 1 genome-wide association in population-based samples (MH081802); Determinants of Adolescent Exercise Behavior (NIDDK R01 DK092127-04); Grand Opportunity grants Integration of Genomics and Transcriptomics (NIMH 1RC2MH089951-01) and Developmental Trajectories of Psychopathology (NIMH 1RC2 MH089995); and the Avera Institute for Human Genetics, Sioux Falls, South Dakota (USA). We also thank the participants and parents for their voluntary participation in the research project of the Netherlands Twin Register. Data were also provided by the developing Human Connectome Project, KCL-Imperial-Oxford Consortium and the work was funded by ERC grant agreement no. 319456, the Wellcome EPSRC Centre for Medical Engineering at Kings College London (WT 203148/ Z/16/Z) and by the National Institute for Health Research (NIHR) Biomedical Research Centre based at Guy's and St Thomas' NHS Foundation Trust and King's College London. We thank all the families who kindly agreed to participate in the project and recognize their particular commitment in remaining engaged with the programme during the COVID-19 pandemic; the Neonatal Intensive Care Unit and the Newborn Imaging Centre at Evelina London Children's Hospital for the support; all the families who contributed with their data; and S. Medland, L. Jiang, E. Hagen, G. Hindley, J. Yang and R. Ma for useful advice at various stages of the research.

## Author contributions

A.G., J.-J.H., E.B.R., M.H.J., F.D., A. Havdahl and A.R. were involved in the conception or design of the work. A.W., V.W., O.A.A., M.B., D.I.B., C.M.M., A.D.E., C.A.H. and A. Havdahl contributed to the data acquisition. A.G., A. Hollowell, E.M.W., M.J.M., L.J.H., E.C.C., V.O., A.W., R.P., H.C., S.W., V.W., O.A.A., M.B., S.J.S. and A.R. were involved in the data preparation and analysis. A.G., A. Hollowell, E.M.W., L.J.H., E.C.C., R.P., H.C., S.W., E.ME., B.S.P., T.A., F.D., S.J.S., A. Havdahl and A.R. interpreted the data. A.G., A. Hollowell, E.M.W., M.J.M. and A.R. drafted the manuscript. All authors reviewed and revised the manuscript.

## Competing interests

O.A.A. is a consultant to cortechs.ai and Precision Health, and receives speaker honoraria from Janssen, Lundbeck, Sunovion, Lilly, and Otsuka. S.J.S. receives research funding from BioMarin Pharmaceutical. The other authors declare no competing interests.

## Additional information

**Extended data** is available for this paper at https://doi.org/10.1038/s41562-025-02145-1.

**Correspondence and requests for materials** should be addressed to Angelica Ronald.

[1]Department of Psychology, University of Essex, Wivenhoe Park, Colchester, UK. [2]Centre for Brain and Cognitive Development, Department of Psychological Sciences, Birkbeck University of London, London, UK. [3]Institute of Developmental and Regenerative Medicine, Department of Paediatrics, University of Oxford, Oxford, UK. [4]School of Psychology, Faculty of Health and Medical Sciences, University of Surrey, Guildford, Surrey, UK. [5]Research Department, Lovisenberg Diaconal Hospital, Oslo, Norway. [6]PsychGen Centre for Genetic Epidemiology and Mental Health, Norwegian Institute of Public Health, Oslo, Norway. [7]Population Health Sciences, Bristol Medical School, University of Bristol, Bristol, UK. [8]Department of Biological Psychology, Faculty of Behavioral and Movement Sciences, Vrije Universiteit Amsterdam, Amsterdam, the Netherlands. [9]Department of Psychiatry, University Medical Center of Groningen, University of Groningen, Groningen, the Netherlands. [10]MRC Unit for Lifelong Health and Ageing at UCL, University College London, London, UK. [11]Research Department of Early Life Imaging, School of Biomedical Engineering and Imaging Sciences, King's College London, London, UK. [12]Department of Medical and Molecular Genetics, School of Basic and Medical Biosciences, King's College London, London, UK. [13]Fetal-Neonatal Neuroimaging and Developmental Science Center, Boston Children's Hospital, Boston, MA, USA. [14]Division of Newborn Medicine, Harvard Medical School, Boston, MA, USA. [15]Department of Psychiatry and Psychology, University of Cambridge, Cambridge, UK. [16]PROMENTA Research Center, University of Oslo, Oslo, Norway. [17]Centre for Precision Psychiatry, Institute of Clinical Medicine, University of Oslo and Division of Mental Health and Addiction, Oslo University Hospital, Oslo, Norway. [18]KG Jebsen Centre for Neurodevelopmental disorders, University of Oslo, Oslo, Norway. [19]Department of Child and Youth Psychiatry and Psychology, Amsterdam Reproduction and Development Research Institute, Amsterdam Public Health Research Institute, Amsterdam UMC, Amsterdam, the Netherlands. [20]Arkin Mental Health Care, Amsterdam, the Netherlands. [21]Levvel, Academic Center for Child and Adolescent Psychiatry, Amsterdam, the Netherlands. [22]Child Health Research Centre, University of Queensland, Brisbane, Australia. [23]Child and Youth Mental Health Service, Children's Health Queensland Hospital and Health Service, Brisbane, Australia. [24]Max Planck Institute for Psycholinguistics, Nijmegen, the Netherlands. [25]MRC Integrative Epidemiology Unit, University of Bristol, Bristol, UK. [26]Donders Institute for Brain, Cognition and Behaviour,

Radboud University, Nijmegen, the Netherlands. [27]Department of Complex Trait Genetics, Center for Neurogenomics and Cognitive Research, Vrije Universiteit, Amsterdam, the Netherlands. [28]University Medical Center Psychopathology and Emotion Regulation (ICPE), Department of Psychiatry, University Medical Center Groningen, University of Groningen, Groningen, the Netherlands. [29]Broad Institute, Boston, MA, USA. [30]Department of Psychology, University of Cambridge, Cambridge, UK. [31]Department of Population Health Sciences, University of Leicester, Leicester, UK. [32]Department of Psychiatry and Behavioral Sciences, UCSF Weill Institute for Neurosciences, University of California, San Francisco, CA, USA. [33]These authors contributed equally: Stephan J. Sanders, Alexandra Havdahl, Angelica Ronald. ✉e-mail: a.ronald@surrey.ac.uk

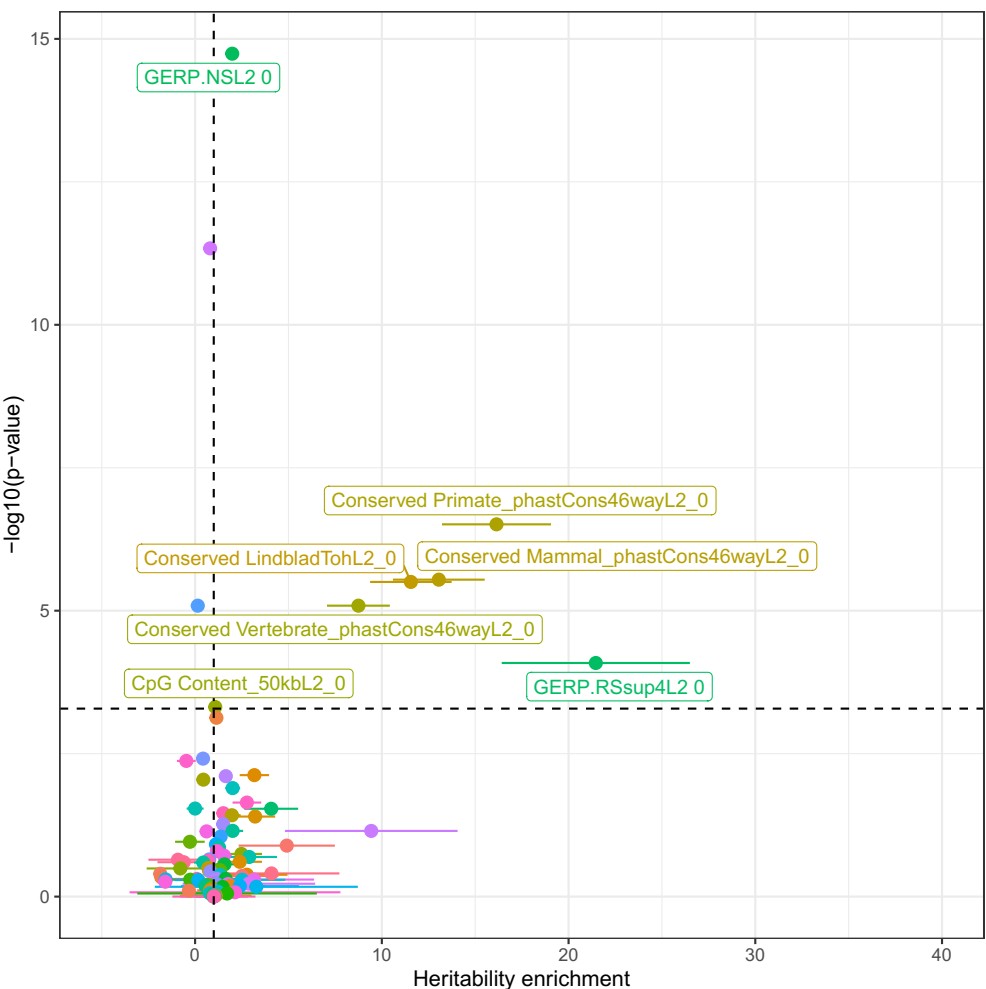

**Extended Data Fig. 1 | Partitioned heritability enrichment by functional annotation.** Enrichment of age at onset of walking GWAS signal by functional genomic annotation. Points represent the heritability enrichment estimate +/− standard errors of the enrichment estimates, obtained in LDSC[40] (two-sided test). The dashed horizontal line represents statistical significance based on Bonferroni correction for multiple testing (Supplementary Table 12). Genomic annotations with significant enrichment for age at onset of walking are labelled. Dots are colored using a spectrum of colors based on alphabetical order.

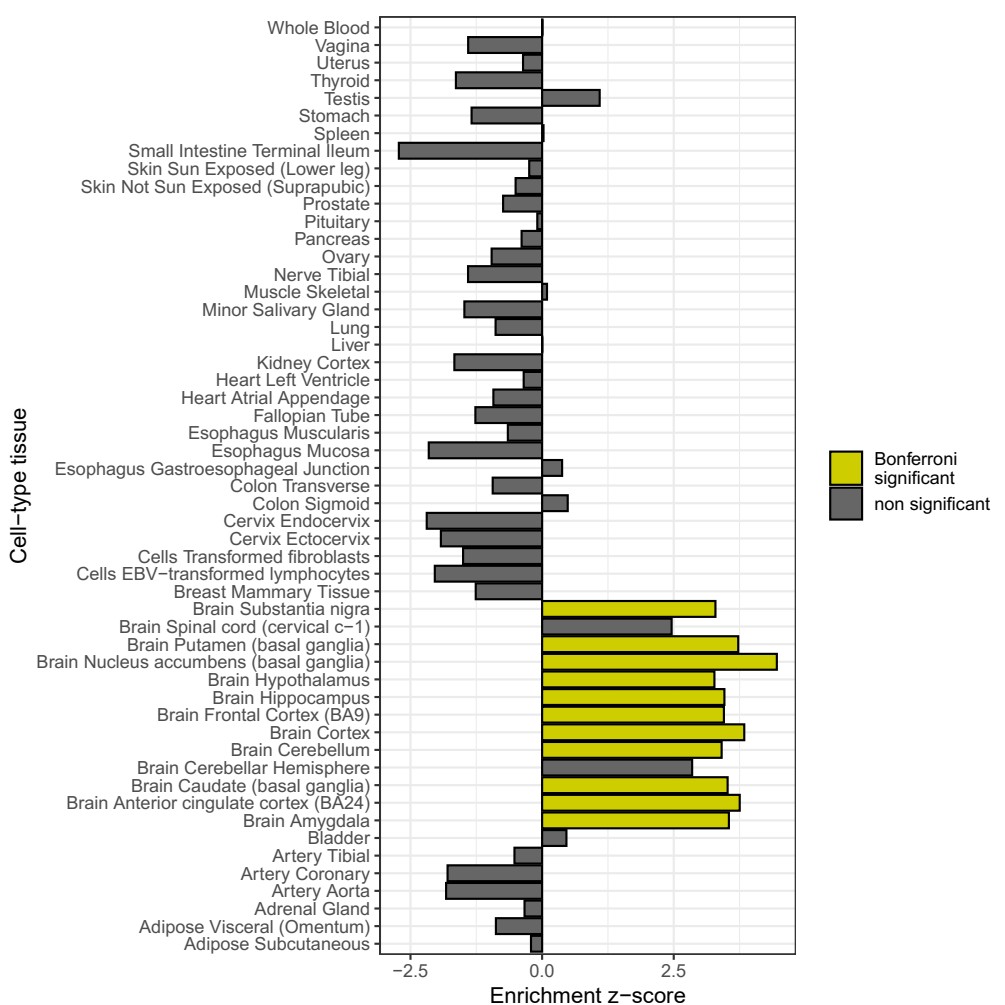

**Extended Data Fig. 2 | Partitioned heritability enrichment by cell type.** Tissue enrichment based on LDSC partitioned heritability analysis[41]. Statistically significant enrichments after correcting two-sided p-values for multiple comparisons using the Bonferroni method are highlighted as yellow bars.

# Reporting Summary

## Statistics

For all statistical analyses, confirm that the following items are present in the figure legend, table legend, main text, or Methods section.

| n/a | Confirmed | |
|-----|-----------|---|
| ☐ | ☒ | The exact sample size (*n*) for each experimental group/condition, given as a discrete number and unit of measurement |
| ☐ | ☒ | A statement on whether measurements were taken from distinct samples or whether the same sample was measured repeatedly |
| ☐ | ☒ | The statistical test(s) used AND whether they are one- or two-sided<br>*Only common tests should be described solely by name; describe more complex techniques in the Methods section.* |
| ☐ | ☒ | A description of all covariates tested |
| ☐ | ☒ | A description of any assumptions or corrections, such as tests of normality and adjustment for multiple comparisons |
| ☐ | ☒ | A full description of the statistical parameters including central tendency (e.g. means) or other basic estimates (e.g. regression coefficient) AND variation (e.g. standard deviation) or associated estimates of uncertainty (e.g. confidence intervals) |
| ☐ | ☒ | For null hypothesis testing, the test statistic (e.g. *F*, *t*, *r*) with confidence intervals, effect sizes, degrees of freedom and *P* value noted<br>*Give P values as exact values whenever suitable.* |
| ☐ | ☒ | For Bayesian analysis, information on the choice of priors and Markov chain Monte Carlo settings |
| ☒ | ☐ | For hierarchical and complex designs, identification of the appropriate level for tests and full reporting of outcomes |
| ☐ | ☒ | Estimates of effect sizes (e.g. Cohen's *d*, Pearson's *r*), indicating how they were calculated |

*Our web collection on statistics for biologists contains articles on many of the points above.*

## Software and code

Policy information about availability of computer code

| Data collection | No software was used for data collection for the purpose of this study. |
|-----------------|-------------------------------------------------------------------------|
| Data analysis | MoBa phenotype data analysis was performed using the R package phenotools: https://github.com/psychgen/phenotools<br>Genotype data analysis was performed in PLINK 1.9 https://www.cog-genomics.org/plink/ , PLINK 2 https://www.cog-genomics.org/plink/2.0/, GCTA https://yanglab.westlake.edu.cn/software/gcta/#Overview, R. Summary statistics quality control was performed using the R package GWASinspector http://gwasinspector.com/ . The meta-analysis was performed in METAL https://genome.sph.umich.edu/wiki/METAL. Post-GWAS fine mapping and functional annotation was performed in FUMA (version 1.5.2) and MAGMA (version 1.08) https://fuma.ctglab.nl/ . Colocalization analyses were conducted using R libraries locuscomparer 1.0.0 (https://github.com/boxiangliu/locuscomparer), coloc 5.2.2 (https://CRAN.R-project.org/package=coloc ) and susieR 0.12.35 (https://github.com/stephenslab/susieR ). SNP heritability, genetic correlation and partitioned heritability were estimated using LD score regression https:/ /github.com/bulik/ldsc . Genomic SEM is available here: https:/ /github.com/GenomicSEM/GenomicSEM . The MiXeR program is available here: https:/ /github.com/precimed/mixer. The polygenic scores were calculated using PRS-cs https:/ /github.com/getian107 /PRScs . The scripts for the within and between-family polygenic score analyses are available here: https:/ /github.com/PerlineDemange/GeneticNurtureNonCog/. |

For manuscripts utilizing custom algorithms or software that are central to the research but not yet described in published literature, software must be made available to editors and reviewers. We strongly encourage code deposition in a community repository (e.g. GitHub). See the Nature Portfolio guidelines for submitting code & software for further information.

# Data

Policy information about availability of data

All manuscripts must include a data availability statement. This statement should provide the following information, where applicable:

- Accession codes, unique identifiers, or web links for publicly available datasets
- A description of any restrictions on data availability
- For clinical datasets or third party data, please ensure that the statement adheres to our policy

The summary statistics of the genome-wide association study of age at onset of walking are available on FigShare (doi: 10.6084/m9.figshare.28071566 ).

eQTL results for the ROSMAP, Mayo TCX, Mayo CER and cortical meta-analysis from Sieberts et al are available through the AMP-AD Knowledge Portal: https://www.synapse.org/Synapse:syn2580853/wiki/409840

The accession number for the raw RNA-seq and WGS data from BrainVar, along with processed files, is PsychENCODE Knowledge Portal: syn21557948 on Synapse.org (https://www.synapse.org/#!Synapse:syn4921369).

Developing Human Connectome project data is open-access and data are available for download via https://nda.nih.gov/edit_collection.html?id=3955.

# Research involving human participants, their data, or biological material

Policy information about studies with human participants or human data. See also policy information about sex, gender (identity/presentation), and sexual orientation and race, ethnicity and racism.

| | |
|---|---|
| Reporting on sex and gender | Biological sex was used as a co-variate and for sex-stratified analyses in this study. Sex was determined based on sex chromosome information and self or parent-report. |
| Reporting on race, ethnicity, or other socially relevant groupings | Genetic ancestry was estimated based on principal component analysis on the cohorts' genotype data following standard procedures (e.g., Marees et al., 2018 doi: 10.1002/mpr.1608, Corefield et al., 2022 doi: 10.1101/2022.06.23.496289). |
| Population characteristics | The participants were 70,560 European-ancestry genotyped children (34,345 males and 33,623 females) from four cohorts: Norwegian Mother, Father and Child Cohort Study (MoBa, N = 58,302), Netherlands Twin Register (NTR, N = 6,251), Lifelines multi-generational prospective population-based birth cohort study (N = 3,415) and Medical Research Council National Study for Health and Development (NSHD, N = 2,592). Genotype data were collected at birth (MoBa), at multiple time points (NTR), during one of the visits for data collection (Lifelines), no information as to when it was collected is provided for the NSHD cohort. Phenotype data were parent-report questionnaires collected when the children were 18 to 36 months for MoBa, after the second birthday of the NTR children (mean age 2.34 years, SD 0.25), during the first assessment of participants, known as the baseline assessment for Lifelines, at age 2 years of the children for NSHD. The Developing Human Connectome Project included newborn infants born in London (UK) across a spread of gestational ages at birth (range: 23 to 43 + 1 weeks + days) and post-menstrual ages at the time of study (range: 26 + 5 to 45 + 1).The subsample included in the current study comprised 264 term-born infants (137 male, 127 female) with available genotype and T2 magnetic resonance images. |
| Recruitment | For MoBa, participants were recruited through hospitals, firstly in Bergen starting in 1999, and then expanding to 50 of 52 Norway's hospitals with maternity units. NTR is a population-based cohort of over 200,000 people from across the Netherlands. It consists of twin-families, i.e. twins, their parents, spouses and siblings aged between O and 99 years at recruitment. NTR started around 1987 with new-born twins and adolescent and adult twins. The Lifelines study was established in 2006 and asked all the general medical practitioners operating in the provinces of Friesland, Groningen and Drenthe to invite their patients aged between 25 and 50 to take part in the study unless the patient met one of five exclusion criteria (as determined by the practitioner): a) having a severe psychiatric or physical illness (such that the individual was not fully capable to make rational decisions), b) life expectancy of less than 5 years, c) being unable to complete a Dutch language questionnaire, d) a lack of ability in the Dutch language, or e) not being able to visit their medical practitioner (http://wiki-lifelines.web.rug.nl/doku.php?id=cohort). NSHD recruited all women who gave birth in a single week in March 1946 in England, Wales or Scotland. Infants in the Developing Human Connectome Project were recruited at St Thomas' Hospital, London and imaged at the Evelina Newborn Imaging Centre, Centre for the Developing Brain, King's College London, United Kingdom. Pregnant woman with fetal age estimated from last menstrual period and live infants between 23 and 44 weeks gestational age were invited to the study. Mothers or infants with contraindication to Magnetic Resonance imaging, preterm infants who are too unwell to tolerate the scanning period, and language difficulties preventing proper communication about the trial and the consent process. |
| Ethics oversight | This study and the related secondary data analysis were approved by the Departmental Ethics Committee of the Psychological Science Department of Birkbeck, University of London on 27th October 2020 (reference number 2021007). MoBa and the related data collection was authorised by a licence from the Norwegian Data Protection Agency and an approval from the The Regional Committees for Medical and Health Research Ethics (REK). MoBa is regulated by the Norwegian Health Registry Act. The current study was approved by REK (2016/1702). Informed consent for NTR was obtained from parents or guardians. The study was approved by the Central Ethics Committee on Research Involving Human Subjects of the VU University Medical Centre, Amsterdam, an Institutional Review Board certified by the U.S. Office of Human Research Protections (IRB number IRB00002991 under Federal-wide Assurance-FWA00017598; IRB/institute codes, NTR 03-180). |

For Lifelines, participants in Lifelines gave written consent prior to physical examination. The study is conducted according to the principles of the Declaration of Helsinki and in accordance with the UMCG research code and is approved by the medical ethical committee of UMCG (document number METC UMCG METc 2007/152).
For NSHD, the collection of blood samples and DNA information from the participants was approved by ethical approval reference MREC no. 98/2/121.
The Developing Human Connectome Project (dHCP) was approved by the UK Health Research Authority (Research Ethics Committee reference number: 14/LO/1169) and written parental consent was obtained in every case for imaging and open data release of the anonymized data.

Note that full information on the approval of the study protocol must also be provided in the manuscript.

# Field-specific reporting

Please select the one below that is the best fit for your research. If you are not sure, read the appropriate sections before making your selection.

☒ Life sciences          ☐ Behavioural & social sciences          ☐ Ecological, evolutionary & environmental sciences

For a reference copy of the document with all sections, see nature.com/documents/nr-reporting-summary-flat.pdf

# Life sciences study design

All studies must disclose on these points even when the disclosure is negative.

| | |
|---|---|
| Sample size | Cohorts were invited to take part in the GWAS meta-analysis if they had available phenotype data (age at onset of independent walking in months) and genotype for more than 1,000 individuals. The minimum sample size for participating in the GWAS meta-analysis was defined a priori and pre-registered on OSF (https://osf.io/jyk6d/ ).<br>The total sample size for this GWAS meta-analysis was 70,560 infants. With 2,525 SNPs (11 independent loci) passing a genome-wide significance threshold, the current sample demonstrated enough power to detect genetic variation associated with age at onset of walking with $p < 5 \times 10\text{-}8$. |
| Data exclusions | In all the individual cohorts, samples were excluded from the GWAS if they had missing phenotype data, if they presented excess autosomal heterozygosity, mismatch between self-reported and genetic sex, XXV genotype and other aneuploidies, individual genotyping rate< 90%. Duplicate samples and samples whose genetically determined ancestry did not overlay with the European ancestry cluster based on a reference panel were also excluded. |
| Replication | For colocalization, results were attempted twice: once in cerebellum and once in cortex. Results in RBL2 were replicated in both, and we reported differences in results at genomic locus 6 which has a complex haplotype structure.<br><br>Replication of the association between polygenic score (PGS) and infant gross-motor skills was performed on the Developing Human Connectome Project (dHCP) cohort, including 217 European term-born infants that had been assessed using the Bayley-III Scales of Infant and Toddler Development at 18-months of age. The relationship between scaled gross motor score and the age at onset of walking PGS was tested using a regression model. Sex, gestational age at birth, birth weight z-score, home environment score (as a proxy for socioeconomic status) and 10 ancestral PCs were included as covariates. We confirmed in this independent sample that the age at onset of walking PGS was significantly associated with lower Bayley's gross motor score, indicating worse/possibly delayed gross motor skills ($\beta = -0.161$, SE= 0.070, p = 0.022). |
| Randomization | There was no randomization involved in this study. This was not a randomized control trial, but an observational design. |
| Blinding | There was no blinding involved in this study with respect to group allocation, as this was an observational study. |

# Reporting for specific materials, systems and methods

We require information from authors about some types of materials, experimental systems and methods used in many studies. Here, indicate whether each material, system or method listed is relevant to your study. If you are not sure if a list item applies to your research, read the appropriate section before selecting a response.

## Materials & experimental systems

| n/a | Involved in the study |
|---|---|
| ☒ | ☐ Antibodies |
| ☒ | ☐ Eukaryotic cell lines |
| ☒ | ☐ Palaeontology and archaeology |
| ☒ | ☐ Animals and other organisms |
| ☒ | ☐ Clinical data |
| ☒ | ☐ Dual use research of concern |
| ☒ | ☐ Plants |

## Methods

| n/a | Involved in the study |
|---|---|
| ☒ | ☐ ChIP-seq |
| ☒ | ☐ Flow cytometry |
| ☐ | ☒ MRI-based neuroimaging |

# Plants

| | |
|---|---|
| Seed stocks | N/A |
| Novel plant genotypes | N/A |
| Authentication | N/A |

# Magnetic resonance imaging

## Experimental design

| | |
|---|---|
| Design type | N/A |
| Design specifications | N/A |
| Behavioral performance measures | N/A |

## Acquisition

| | |
|---|---|
| Imaging type(s) | structural |
| Field strength | 3T |
| Sequence & imaging parameters | T2-weighted images were obtained using a Turbo Spin Echo sequence, acquired in two stacks of 2D slices (in sagittal and axial planes), using parameters: TR=12s, TE=156ms, SENSE factor 2.11 (axial) and 2.58 (sagittal) with overlapping slices (resolution 0.8x0.8xl.6mm3). |
| Area of acquisition | whole brain scan |
| Diffusion MRI | ☐ Used   ☒ Not used |

## Preprocessing

| | |
|---|---|
| Preprocessing software | Advanced Neuroimaging Tools (ANTs) |
| Normalization | The log-Jacobian determinant images were calculated by applying ANTs algorithms to the non-linear transformation deformation tensor fields. |
| Normalization template | 40-week dHCP neonatal atlas (https://brain-development.org/brain-atlases/atlases-from-the-dhcp-project/) |
| Noise and artifact removal | *Describe your procedure(s) for artifact and structured noise removal, specifying motion parameters, tissue signals and physiological signals (heart rate, respiration).* |
| Volume censoring | *Define your software and/or method and criteria for volume censoring, and state the extent of such censoring.* |

## Statistical modeling & inference

| | |
|---|---|
| Model type and settings | Permutation testing using the randomise function, part of the FMRIB Software Library (FSL) was used with a general linear model. |
| Effect(s) tested | beta estimate of a linear regression. |
| Specify type of analysis: | ☐ Whole brain   ☒ ROI-based   ☐ Both |
| Anatomical location(s) | *Describe how anatomical locations were determined (e.g. specify whether automated labeling algorithms or probabilistic atlases were used).* |
| Statistic type for inference | voxel-wise |

(See Eklund et al. 2016)

| Correction | Threshold-Free Cluster Enhancement (TFCE) and Family-Wise Error (FWE) rate were applied to correct for multiple comparisons between voxels. |

## Models & analysis

| n/a | Involved in the study |
|---|---|
| ☒ ☐ | Functional and/or effective connectivity |
| ☒ ☐ | Graph analysis |
| ☐ ☒ | Multivariate modeling or predictive analysis |

| Multivariate modeling and predictive analysis | Independent variable: polygenic score. Covariates: gestational age, postmenstrual age at scan, sex, weight-z-score and 10 ancestral principal components as covariates |

