## [Peer Review File · Nature Human Behaviour]

Genome-wide association meta-analysis of age at onset of walking in over 70,000 infants of European ancestry

Corresponding Author: Professor Angelica Ronald

This manuscript has been previously reviewed at another journal. This document only contains information relating to versions considered at Nature Human Behaviour.

Version 0:

Decision Letter:

Our ref: NATHUMBEHAV-24093766-T

12th November 2024

Dear Dr. Ronald,

Thank you for submitting your revised manuscript "Genome-wide association meta-analysis of age at onset of walking" (NATHUMBEHAV-24093766-T). It has now been seen by two of the original referees and their comments are below (Reviewer 1 was Reviewer 3 in the previous round of review; Reviewer 2 was also Reviewer 2 previously). As you can see, the reviewers find that the paper has improved in revision. We will therefore be happy in principle to publish it in Nature Human Behaviour, pending minor revisions to satisfy the referees' final requests and to comply with our editorial and formatting guidelines.

Specifically, we ask that you address all comments made by Reviewer 2, except their request for LAVA analyses, which we agree are best presented in a follow-up publication.

We are now performing detailed checks on your paper and will send you a checklist detailing our editorial and formatting requirements within two weeks. Please do not upload the final materials and make any revisions until you receive this additional information from us.

Sincerely,

Nature Human Behaviour

Reviewer #1 (Reviewer #3 in previous round) (Remarks to the Author):

The authors have thoroughly addressed all the previous concerns from Reviewer #3 and Reviewer #4. The authors were exceedingly responsive to the Reviewers' requests for additional data analyses. They have included additional data (where needed) updated results and conclusions accordingly, clarified methods, and acknowledged additional limitations. I look forward to seeing these research findings published.

Reviewer #2 (Remarks to the Author):

Review: 31495 Nat Hum Beh – meta-analysis GWAS of age at onset of walking

General:

This study concerns a genome-wide meta-analysis of 4 cohorts of age-at-onset-of-walking. The analyses are quite comprehensive and the study is interesting. I do have a few comments.

Lines 164-173: This summary of results seems a bit out of place: I would expect it in the abstract or the discussion but not here as such?

Line 197 and on: I believe it is customary to not only mention the number of independent loci but also the overall number of genome-wide significant SNPs?

Lines 232-236: I like the male/female distinction. Is it really the case that h^2_{SNP} was 23.06 for both yet 24.13 combined?

Line 242-243: I would mention the h^2_{SNP} per sample and the rgs between samples (Suppl note C) at the start of this paragraph as it is informative on whether we should meta-analyse these cohorts at all?

Fig 1: legend says inflation factor = 1.27, while the text says 1.23 (line 187) and LDSC intercept of 1.00 while the text says 1.008 (line 188; rounded would be 1.01).

Line 238: sentence doesn't run properly.

Line 269: inconsistent use of . and , in larger numbers (please check throughout: I also saw instances in the supplemental tables).

Line 277: sentence doesn't run properly ("we found THAT...")

Line 409: the p-value of .70 does not seem to fit the chi-sq value of .04 (in R: 1-pchisq(.04,1) gives .84).

Line 411: seems to fit more with the previous paragraph.

Line 451: "of AOW" I think, not "with AOW".

Lines 475-486: what about ADHD? It seems to odd to single out ADHD in the previous paragraph (451-459) and not discuss it here? Also: the SNP-overlap with educ attainment and cognition seems very high and the discordance signal quite substantial: would it make sense to run LAVA on these traits and see whether maybe some local correlations are negative, some positive, cancelling each other out in the global LDSC rg? (609-611 also notes explicitly that these discordant effects are of interest: LAVA allows one to study these discordant effects specifically).

Line 566: delete "that" ("2 of which that colocalised").

Line 596 "our third conclusion from our results": unfortunately phrased. I find "our xx conclusion" somewhat less formal anyways.

Lines 598 and on: be careful not to interpret or phrase these genetic correlations as if they were phenotypic.

Line 613 -619: genetic correlations interpreted very much in phenotypic terms, please consider careful rephrasing.

Line 627: I believe previously this rg was -.18, not -.19?

Lines 642-643: "that for children with genetically-influenced protracted subcortical neurogenesis in the prenatal period"; are there also children without genetically-influenced subcortical neurogenesis?

General: I see, 1, 2 and 3 decimals being used: I would opt for 2 throughout, not a mix?

Suppl Table 1:

there seem to be considerable mean differences in AOW between the 4 samples; can these be explained?

Suppl Table 2:

* inconsistent use of . and , in larger numbers.

* the sample sizes before QC are much much larger, especially for cohort 1, 2 and 4: I presume participants dropped out because they did not have phenotypic information of AOW and not all because of genetic QC reasons? I would suggest to report the N with valid genotype/phenotype data in that first column, and not all people in the database (if that is what the authors did: not sure really).

Sophie van der Sluis

Version 1:

Decision Letter:

Dear Professor Ronald,

We are pleased to inform you that your Article "Genome-wide association meta-analysis of age at onset of walking in over 70,000 infants of European ancestry", has now been accepted for publication in Nature Human Behaviour.

Authors may need to take specific actions to achieve [compliance with funder and institutional open access mandates](https://www.springernature.com/gp/open-research/funding/policy-compliance-faqs). If your research is supported by a funder that requires immediate open access (e.g. according to [Plan S principles](https://www.springernature.com/gp/open-research/plan-s-compliance)) then you should select the gold OA route, and we will direct you to the compliant route where possible. For authors selecting the subscription publication route, the journal's standard licensing terms will need to be accepted, including [self-archiving policies](https://www.springernature.com/gp/open-research/policies/journal-policies). Those licensing terms will supersede any other terms that the author or any third party may assert apply to any version of the manuscript.

With best regards,

Nature Human Behaviour

P.S. Click on the following link if you would like to recommend Nature Human Behaviour to your librarian <http://www.nature.com/subscriptions/recommend.html#forms>

** Visit the Springer Nature Editorial and Publishing website at http://editorial-jobs.springernature.com?utm_source=ejP_NHumB_email&utm_medium=ejP_NHumB_email&utm_campaign=ejp_NHumB for more information about our career opportunities. If you have any questions please click [here](mailto:editorial.publishing.jobs@springernature.com).

Open Access This Peer Review File is licensed under a Creative Commons Attribution 4.0 International License, which permits use, sharing, adaptation, distribution and reproduction in any medium or format, as long as you give appropriate credit to the original author(s) and the source, provide a link to the Creative Commons license, and indicate if changes were made. In cases where reviewers are anonymous, credit should be given to 'Anonymous Referee' and the source. The images or other third party material in this Peer Review File are included in the article's Creative Commons license, unless indicated otherwise in a credit line to the material. If material is not included in the article's Creative Commons license and your intended use is not permitted by statutory regulation or exceeds the permitted use, you will need to obtain permission directly from the copyright holder.

Dear Professor Ronald,

Please accept my apologies for the protracted review process.

Your Article entitled "Genome-wide association meta-analysis of age at onset of walking" has now been seen by 4 referees, whose comments are attached. In the light of their advice we have decided that we cannot offer to publish your manuscript in Nature Medicine.

While the referees find your work of some interest, they raise concerns about technical issues and clinical advance. We feel that these reservations are sufficiently important as to preclude publication of this study in Nature Medicine.

Although we cannot offer to publish your manuscript, I suggest that you consider one of our sister journals, either Nature Genetics or Nature Communications, as a suitable venue for a revised version of this work. Please note that I haven't discussed the manuscript with the editors from either of these journals, but if you would like me to, I would be happy to do so, in which case please let me know. To transfer your manuscript, please use our manuscript transfer portal. You will not have to re-supply manuscript metadata and files, unless you wish to make modifications. For more information, please see our manuscript transfer FAQ page. For more information, please see our manuscript transfer FAQ page.

I am sorry that we cannot be more positive on this occasion but hope that you will find our referees' comments helpful when preparing your paper for submission elsewhere.

Sincerely,

██████████

██████████

Nature Medicine

Reviewers' Comments:

Reviewers' Comments:

Reviewer #1:

Remarks to the Author:

This is well written paper on age at onset of walking. With my limited competence in the details of the statistical models used, it seems to me that they have done exactly

what is expected to as good as possible unravel the genetic architecture behind the phenotype, and provide new biological insights which might be of importance for future research.

Many thanks to Reviewer 1 for their encouraging feedback.

Reviewer #2:

Remarks to the Author:

Review NatMed-163318

Topic: genetic meta-analysis of age-at-onset of walking

This study concerns a large GWAS on age at onset of walking, linking the results to specific genes and other phenotypes. As age at onset of walking is considered a relevant indicator in early childhood, the study on this phenotype is relevant and the genetic approach makes sense as the phenotype has been shown to be heritable. Methods are appropriate, but could maybe be extended somewhat (see suggestions below).

This is an interesting and well executed study. I do have some questions/suggestions.

Intro lines 174-188: summary of the results: quite detailed. It is very similar to what we can read in the abstract. Maybe consider shortening this section. I find the overall introduction informative but rather lengthy.

We have now shortened the introduction, including the last paragraph. We removed some of the repetition of text provided in the abstract, as follow (page 4):

“Here we present the first genome-wide association study (GWAS) meta-analysis of AOW in a sample of 70,560 children from four European-ancestry cohorts. First, we found AOW to be a highly polygenic trait with 24% SNP heritability. We identified 11 independent genetic loci, one of which colocalized with expression quantitative trait loci (eQTLs) in *RBL2*. This gene is associated with a rare developmental disorder that involves delayed or absent walking. AOW showed significant genetic correlations with physical health indicators, neurodevelopmental conditions and cognitive traits, psychiatric disorders and cortical phenotypes. The polygenic score showed out-of-sample prediction of 3-5.6% and was positively associated with the volume of neonatal brain structures involved in motor function”.

“age at onset of walking” appears over 70 times in the document: rather than writing it out, the authors could consider an abbreviation (which would save at least 4*70 words!).

Thank you for this suggestion. We now abbreviated “age at onset of walking” with “AOW” throughout the document.

Lines 199-202: for 2 samples, the country of origin is clear; maybe add that information also explicitly for LifeLines and NSHD?

We have now added country of origin information on page 5:

“We conducted a GWAS meta-analysis of AOW in 70,560 children including data from four European-ancestry cohorts: Norwegian Mother, Father and Child Cohort Study^{21,22} (MoBa, N = 58,302), Netherlands Twin Register²³ (NTR, N = 6,251), Lifelines multi-generational prospective population-based birth cohort study²⁴ from the North of the Netherlands (N = 3,415) and United Kingdom Medical Research Council National Study for Health and Development²⁵ (NSHD, N = 2,592).”

Lines 204-205: the inflation factor of the Norwegian sample is >1.10 and the authors reason this is due to high polygenicity. However, assuming that the trait is not less polygenetic in the other 3 samples, why do we then not see this inflation in the other samples? There, thus, must also be something else going on. Is it maybe the much larger sample size of the Norwegian sample that plays a role too? There are other differences between the samples too that might affect the genetic signal (eg phenotypic mean MOBA is quite a bit lower than the other samples, phenotypic range is also much wider in the Norwegian sample). The number of SNPs in MOBA prior to QC is also much much larger than for the other samples, for which the eventual SNP sample sizes are smaller and where imputation has a much larger effect on the final sample sizes.

Thank you for the opportunity to clarify the relationship between inflation factor, SNP effects and sample sizes in our study. Indeed, as the reviewer points out the inflation factor relates to the sample size (see Yang et al., 2011, <https://doi.org/10.1038/ejhg.2011.39>).

The Jang et al. study also demonstrated that lambda GC does not change as a function of the number of included SNPs (see Figure 2f), therefore the number of initial SNPs is not expected to contribute to the pattern of inflation. This is confirmed by the results of our inflation checks in MoBa (summarised in Supplementary Table S1) where we did not find the same inflation pattern in the same sample when looking at another infant motor phenotype, namely fine motor skills at 18 months (lambda GC = 1.040).

To test whether the range in phenotype values played a role with respect to the observed inflation, we calculated the lambda GC in the MoBa sample after removing individuals with a phenotype value that exceeded the range observed in the other studies (i.e., 25 months), however the lambda GC remained unchanged (Supplementary Note A, Supplementary Table SM1).

Our quality control checks performed in GWASinspector indicate the effect sizes per SNPs included in the individual GWAS are distributed around zero for MoBa as well as for the

other cohorts, with standard errors widening as a function of the sample size. This indicates that results in MoBa are not biased towards larger effects.

Precision by Sample Size

High quality SNPs only!

We now mention the relationship between inflation and sample size in the main text (page 5) to make it readily accessible to the readers.

“The observed inflation is likely explained by trait polygenicity (Linkage Disequilibrium Score Regression, LDSC, intercept = 1.008 (0.008)^{26,27}, see Supplementary Note A for a detailed investigation of the observed inflation). Furthermore, the other **smaller** cohorts’ inflation factors were below the recommended threshold of 1.10 (NTR $\lambda_{GC} = 0.975$, Fig. S4, Lifelines $\lambda_{GC} = 1.001$, Fig. S6, NSHD $\lambda_{GC} = 1.002$, Fig. S8), which is expected given the positive relationship between inflation and sample size²⁸.”

How many SNPs were overlapping between the 4 samples? A meta-analysis on 4 samples of which SNPs do not overlap creates the difficulty that part of the meta-analytic signal may only be based on SNPs from 1 sample, which is especially risky if 1 sample has much more statistical power. Would it be an idea to restrict the SNPs going into the meta-analysis to only those SNPs that were measured (imputed) in all 4 samples? I would expect a Suppl table with results of at least the top X SNPs from the meta-analyses so that the reader can see which samples contributed to the meta-analytic signal but I cant find it among the supplemental tables. Table 1 from the ms could also be supplemented with sample-specific information.

We added the sample size for each of the genomic loci in Table 1. As requested, we also added in a supplementary table (Supplementary Table S5) showing which individual cohorts contributed to the meta-analytic signal for the 11 independent genomic loci in Table 1.

In the *Methods - GWAS meta-analysis* section (page 23), we specify that “only SNPs for which the minimum sample size was 10,000 were retained for further analyses (6,902,401 variants)”. A minimum sample size of 10,000 was obtained if the SNP was available for the MoBa sample, all three of the other cohorts or for all four cohorts. The approach of including in a GWAS meta-analysis the SNPs that are present in a minimum proportion of the entire sample is standard practice in GWAS meta-analyses (see for example the meta-analysis software documentation page:

https://genome.sph.umich.edu/wiki/METAL_Documentation and paper, and other recent meta-analyses of complex human behavioural traits such as Grove et al., 2019 <https://doi.org/10.1038/s41588-019-0344-8> , Demontis et al., 2023 <https://doi.org/10.1038/s41588-022-01285-8>). In GWAS of complex traits, a sample size of >10,000 is required to obtain reliable results (as indicated in Uffelmann et al., 2021, <https://doi.org/10.1038/s43586-021-00056-9>). As such, when we pre-registered the current study, we set this threshold for the minimum sample size to include a SNP in the meta-analysis. Overall, as shown in Table 1, our significant loci were obtained with a sample size higher than 82% of the entire sample, indicating that most individuals contributed to the SNP signal.

We have added reference to Table S5 in the main text, page 5:

“See Table 1 for a full list of significant loci, Supplementary Table S4 for previous associations with complex traits, **and Supplementary Table S5 for which cohorts contributed for each locus**”.

Line 254: “there was no significant heterogeneity” but then in the next 2 lines 2 very small p-values are reported for specific SNPs; what alpha was used here? And what

was the overall meta-analytic I^2 on which the claim in line 254 is based? I miss a supplemental table showing the contribution of each sample to the overall meta-analytic results.

I^2 was computed per SNP and reported as significant at the conventional genome-wide significant threshold of $p = 5 \times 10^{-8}$, as in similar reports (see Demontis et al., 2023, <https://doi.org/10.1038/s41588-022-01285-8>, Supplementary Figure 1). We now added this information in the text on page 8.

We also included an estimate of the M multi-SNP pattern of heterogeneity based on the independent significant SNPs in GWAS meta-analyses, and the M statistic per study in the Supplementary Table S3, as requested.

“There was no **genome-wide** statistically genome-wide significant heterogeneity between cohorts as tested with I^2 **per SNP at the conventional alpha level of $p = 5 \times 10^{-8}$** (maximum $I^2 = 95.3$ for SNPs rs7864115 and rs148684045, $\chi^2(1) = 21.453$ and 21.441 , $p = 3.63 \times 10^{-6}$ and 3.65×10^{-6} , respectively). This indicates that variation of **individual SNP** effects between individual GWASs was **not** due to heterogeneity between the cohorts³³ (Fig. S11, see Supplementary Note C for genetic correlation between cohorts). **Overall, the M multi-SNP heterogeneity metric across the independent lead SNPs³⁴ associated with AOW indicated no systematically more or less influential study (see Supplementary Table 3, all Bonferroni-corrected p s < 0.401).**”

In the Methods section (page 23), we added:

“**The I^2 heterogeneity metric per SNP was calculated in METAL. M multi-SNP heterogeneity statistics indicating whether individual studies were systematically more influential or weaker than average based on their effects was calculated using the getmstatistic R package for the independent lead SNPs (pairwise LD $r^2 < 0.1$, $p < 5 \times 10^{-8}$, N SNPs = 15)³⁴.**”

Lines 290-293: Why is the p-value in line 293 smaller than the p-value in line 290, while the proportion is much larger in line 290?

The chi-squared test is applied here to assess the difference from expected values in a 2x2 contingency table. A higher proportion leads to a greater difference from expectation and therefore a lower p-value. We have added the χ^2 values to the text for clarity, on page 9:

“Using Genomics England PanelApp³⁹, we found that 13 of the 47 genes with Ensembl IDs (27.7%) were associated with intellectual disability (ID, v5.557); this is over double the proportion (2.10 times) of ID-associated genes in the panels as a whole (2,624 out of 19,950, 13.2%; $\chi^2 = 7.45$; $p = 0.006$, two-tailed). These genes include *ATXN2*, *AUTS2*,

CUX2, *FOXP1*, *KANSL1*, and *RBL2* (Table S8). Furthermore, we found 7 of the 47 genes were associated with autism (14.9%), which is over four times the proportion of autism-associated genes in the panel (v0.36, largely based on SFARI-gene⁴⁰) as a whole (734 out of 19,950, 3.68%; $\chi^2 = 13.7$; $p = 0.0002$, two-tailed).”

Here are the two contingency tables.

	GWAS walk	
ID panel	13	34
	2611	17292

	GWAS walk	
ASD panel	7	40
	727	19176

Lines 299-300: “expression of the genes associated with age at onset of walking were significantly enriched”: the expression were significantly enriched: rephrase.

Thank you for pointing out this typo, we have rephrased this sentence as requested on page 9:

“Overall, expression of the genes associated with AOW was significantly enriched between 19 and 24 (late mid-prenatal period) post-conceptual weeks (Fig. S14).”

Line 302-302: Magma gene-set analyses of 10 gene-sets showed no significant results: why these 10 specific gene-sets?

The sentence in the main text (page 9) reads “The MAGMA gene-set analysis yielded no significant results (Table S10).” On Table S9, now renamed as S10 due to the addition of another supplementary table, we originally reported the ten top gene sets, which were all non-significant at an FDR corrected p-value threshold. We showed ten top gene sets to

avoid overloading the readers with more than 17K rows of non-significant results. However, thanks to the reviewer's comment we now understand reporting only the first ten results is confusing and readers might want to see more results, therefore we now report results for all the gene sets with an unadjusted p-value < 0.05.

The caption for Table S10 has been modified as follows:

“Supplementary Table S10. Top curated gene sets and Gene Ontology terms obtained from MsigDB performed in MAGMA using the full distribution of SNP p-values from the age at onset of walking GWAS summary statistics. The Adjusted p-values indicate p-values for enrichment of significant SNPs in gene sets after correction for multiple testing using the Bonferroni method. Gene-sets with an unadjusted p < 0.05 are reported.”

Lines 365-373: The coloc analyses are interesting, but the information in lines 365-373 seems to overlap largely with previous paragraphs: maybe the text can be shortened slightly or the paragraphs can start with “Summarizing,...”.

We have now rephrased this section to remove the part that seemed overlapping, on page 11:

“Group 1 SNPs are only found on one haplotype (dark blue and red, Fig. 2D) resulting in a lower MAF of 30% than the Group 2 SNPs. We infer that one of the Group 1 SNPs has a functional impact above and beyond the decrease in *RBL2* expression mediated by the Group 2 SNPs, to yield the stronger evidence of association with AOW. Annotation of the 125kb locus with VEP⁴⁶ identified rs17800727 as a likely candidate for this effect, since it results in a missense variant (MANE isoform: ENST00000262133.11, p.Tyr210Cys) (Fig. 2E) that is predicted to impact function by some severity metrics (e.g., ‘Damaging’ based on PolyPhen2⁴⁷, CADD⁴⁸ score of 25) but not all (e.g., ‘Tolerated’ based on SIFT). If the missense variant had a loss-of-function effect, it would be on a haplotype that magnifies the functional impact through decreased expression of *RBL2*; future functional studies would be required to validate this impact.”

Figure legend figure 2: for panel B: “The SNPs from ‘A’ are”; maybe rephrase to “panel A” (also later on in that text). Same suggestion holds for later in the figure.

We have rephrased the figure labels as suggested.

“Fig. 2 Colocalization of variants in genomic locus 2. Genomic locus 2 overlaps with a region in which SNPs are predicted to alter *RBL2* expression in the human brain (eQTLs). A. The GWAS evidence for association with age at onset of walking ($-\log_{10}[\text{p-value}]$, y-axis) is plotted against the statistical evidence of being an eQTL for *RBL2* in human adult cerebellum⁴³ ($-\log_{10}[\text{p-value}]$, x-axis) for each SNP (points) within a 2Mb window around

the GWAS peak. Points are colored by linkage disequilibrium (LD) correlation with the lead SNP (*rs17800727*) and these values are used to define two groups. B. The SNPs from **panel A** are shown in the 2Mbp genomic region (x-axis, GRCh37) with protein-coding genes (top), GWAS evidence for association with age at onset ($-\log_{10}[\text{p-value}]$, middle) and statistical evidence for *RBL2* expression in human cerebellum ($-\log_{10}[\text{p-value}]$, y-axis, bottom). Point color matches **panel A**. C. A zoomed in view of the peak indicated by dashed vertical lines in **panel B** shows the GWAS evidence for association with age at onset of walking ($-\log_{10}[\text{p-value}]$, y-axis) by genomic position (x-axis, GRCh37). Color indicates the Minor Allele Frequency (MAF) of each SNP. The locations of protein-coding genes in the region are indicated at the top. A SNP (*rs17800727*) that results in a missense variant (p.Tyr210Cys) in *RBL2* is marked. D. Swarm, violin, and box-plots showing the distribution of *RBL2* expression **in the prefrontal cortex** (transcripts per million (TPM), y-axis). Each point represents the expression of *RBL2* in one of 87 prenatal human cortices (BrainVar⁴⁵) split by genotype into three groups based on zygosity for the Group 2 50% MAF SNPs. The p-value represents the difference between the homozygous alternate and homozygous reference groups (Wilcoxon, two-sided). Bars at the bottom indicate pairs of haplotypes (derived from the data shown in **panel C**, making up each genotype). E. Structure of *RBL2* protein predicted by AlphaFold⁴⁹ with the location of *rs17800727*, p.Tyr210Cys in red⁵⁰.”

I appreciate the information in Figure 3 but it is a very large figure with a lot of white space. Maybe more suited for the supplement, or comprise it somehow?

We have reformatted Fig. 3, as suggested.

Line 438-439: “a preregistered selection”: what was the selection based on? I don’t see a clear explanation in the ms or in the Supplemental files.

In our pre-registration we stated that we were going to investigate the genetic correlation between age at onset of walking and “early physical health indicators, neurodevelopmental conditions and cognitive traits, psychiatric conditions and cortical phenotypes”. On the contrary, the analysis of walk-related phenotypes was not included in our pre-registration. We now made the link to the pre-registration more obvious in the results section of the manuscript on page 13:

“Next, we tested for genetic correlations between **AOW** and a pre-registered selection of physical health, neurodevelopmental, psychiatric, cognitive and cortical phenotypes (<https://doi.org/10.17605/OSF.IO/M2QV3>).”

And at the beginning of the Results section (page 5):

“We conducted a GWAS meta-analysis of **AOW** in 70,560 children including data from four European-ancestry cohorts: Norwegian Mother, Father and Child Cohort Study^{21,22} (MoBa, N = 58,302), Netherlands Twin Register²³ (NTR, N = 6,251), Lifelines multi-generational prospective population-based birth cohort study²⁴ **from the North of the Netherlands** (N = 3,415) and **United Kingdom** Medical Research Council National

Study for Health and Development²⁵ (NSHD, N = 2,592). Analyses were pre-registered on OSF: <https://doi.org/10.17605/OSF.IO/M2QV3>.”

Genetic correlations section (lines 438 and on): The authors now highlight the significant correlations and do not mention the non-significant ones. Please consider discussing the correlations per “theme”; I think this would be more interesting. E.g. “age of onset of walking was genetically correlated with childhood and adulthood BMI but not with birth weight”, and “of the 6 included psychiatric disorders, only ADHD showed a significant genetic correlation with age of onset”.

Thank you for the suggestion, we re-organised the order of the text, as requested, on page 13:

“For physical health, AOW was negatively genetically correlated with childhood BMI⁵² ($r_g = -0.14$, SE = 0.04, 95%CI [-0.22, -0.07]) and adult BMI⁵³ ($r_g = -0.10$, SE = 0.02, 95%CI [-0.14, -0.06]) but not with birth weight. Of the six included psychiatric disorders, ADHD⁵⁴ ($r_g = -0.18$, SE = 0.03, 95%CI [-0.24, -0.12]) and bipolar disorder⁵⁵ ($r_g = 0.07$, SE = 0.02, 95%CI [0.03, 0.12]) showed a significant genetic correlation with AOW. Additionally, AOW was positively genetically correlated with both the cognitive phenotypes, educational attainment⁵⁶ ($r_g = 0.12$, SE = 0.02, 95%CI [0.08, 0.16]) and cognitive performance⁵⁷ ($r_g = 0.09$, SE = 0.03, 95%CI [0.04, 0.14]).

Among thirteen adolescent and adult cortical phenotypes⁵⁸, we observed a significant genetic correlation between AOW and folding index ($r_g = 0.14$, SE = 0.04, 95%CI [0.06, 0.21]), local gyrification index ($r_g = 0.10$, SE = 0.04, 95%CI [0.03, 0.17]) and cortical surface area ($r_g = 0.07$, SE = 0.04, 95%CI [0.002, 0.15]), all of which are measures of cortical expansion, and isotropic volume fraction, that indicates water diffusion in the brain related to ventricles and cerebrospinal fluid ($r_g = 0.10$, SE = 0.04, 95%CI [0.01, 0.17]). There were no significant genetic correlations with the other complex traits tested (see Supplementary Table S13 and Fig. 4A). For motor phenotypes, non pre-registered exploratory analyses showed that AOW was genetically correlated with self-reported walking pace in adults⁵⁹ ($r_g = 0.06$, SE = 0.03, 95%CI [0.01, 0.11]), but not with other motor phenotypes (Table S13).”

Lines 458-463: why would one correct for educational attainment; what is the rationale here? And why only ADHD: what about the other phenotypes whose genetic correlations were initially significant: did they turn insignificant after conditioning on educational attainment?

The ADHD genetic correlation with AOW was the largest magnitude genetic correlation. It has potential implications for understanding ADHD and one point we were curious about is to what extent this genetic correlation between AOW and ADHD was independent of educational attainment genetics, which are known to be linked to ADHD. We thought that

readers would be interested in this point as well, and thus followed up the genetic correlation with ADHD with this GSEM analysis controlling for educational attainment. We found that the genetic correlation between AOW and ADHD was significant after controlling for educational attainment. We appreciate the reviewer's point that it is not pre-registered and we do not systematically run GSEM on all the statistically significant genetic correlations (there are a large number of GSEM analyses we could run following the bivariate genetic correlations). In light of the reviewer's comment, we added additional text to justify this analysis and clarify its exploratory nature (page 14). However, we are happy to take editorial guidance on this and we can delete this additional analysis if necessary.

“The largest magnitude genetic correlation with AOW was with ADHD. In light of the potential implications of this finding, we tested in an exploratory, non-pre-registered analysis, whether the AOW-ADHD genetic correlation remained after controlling the genetic influences of educational attainment, since the latter are also known to be associated with ADHD⁵⁴.”

Lines 472-473: “the BIC values did not support the bivariate models”: what does that imply/what is tested here? I would be explicit about that. Lines 472-480 are somewhat obscure because of this lack of explicitness.

We now added an additional explanation of the model evaluations in the Methods section, on page 26:

“We fitted bivariate models in MiXeR to estimate the genetic overlap that is due to both concordant and discordant SNP effects, between AOW and six other phenotypes that had a Bonferroni-significant genetic correlation with AOW (calculated using LDSC). For each pair of traits, the models were evaluated using differential BIC and AIC values between the ‘best’ bivariate model estimating the optimal amount of polygenic overlap between the two traits (grey areas in Figure 4.B) and two simpler models, namely the ‘minimum’ and the ‘maximum’ overlap models. The ‘minimum’ overlap models used only the minimum number of SNPs to explain the genetic overlap from the LDSC genetic correlation estimate, while the ‘maximum’ overlap models assumed that all the variants associated with the least polygenic of the two traits overlap with the other trait. Positive differential BIC and AIC values indicated the ‘best’ MiXeR bivariate model outperformed the two simpler models. When the summary statistics for the second phenotype in these bivariate analyses came from the case-control GWAS, the N_{eff} was calculated as $4 / (1 / N_{\text{case}} + 1 / N_{\text{control}})$. The MHC (6:26000000-34000000) was excluded from MiXeR analyses due to its complex LD structure, in line with the program recommendations. MiXeR v1.3 was used for these analyses, and the data were prepared using scripts developed by the same group (https://github.com/precimed/python_convert). We considered the bivariate MiXeR model to be supported when the differential AIC value comparing the ‘best’ vs ‘minimal’ model was positive. This criterion ensures

that there is support for the model of the polygenic overlap that includes the added free parameters of this model.”

We also rephrased the sentences in the Results section (page 14) to be more explicit about the meaning of these tests, as requested:

“We applied bivariate mixture modelling to AOW with all other phenotypes with which there was a significant genetic correlation as calculated by LDSC, after correction for multiple testing (as per Fig. 4A). In terms of AIC fit, we found support for the bivariate MiXeR models that estimated the optimal polygenic overlap between AOW and childhood and adult BMI, educational attainment, cognitive performance, ADHD and folding index (see Figure 4B). These models were supported over the ‘minimal model’ which explains the observed LDSC models using the minimal amount of polygenic overlap possible. AIC and BIC values for all correlated phenotypes, and negative log-likelihood plots are provided in Supplementary Table S14 and Supplementary Fig. S18 respectively.”

Lines 482-491: why focus only on the traits showing most overlap? Is it not particularly interesting to find significant genetic correlations with traits that apparently show (very) little overlap? Also: the authors mention quite some sign-discordance between age of onset of walking and EA/IQ, respectively: what can we conclude from this/ how can this be made relevant?

We now included in the main text reference to the traits with minimal overlap (page 14):

“In terms of the proportion of the SNPs contributing to the polygenicity of AOW that overlap with other phenotypes investigated, the traits investigated that showed the most overlap were educational attainment and cognitive performance (82.44% and 91.07% respectively). Of these overlapping SNPs, the fractions of SNPs that had concordant directions of effect were 55.10% and 53.70% for educational attainment and cognitive performance respectively. On the contrary, little SNP overlap, despite significant genetic correlation, was found with childhood BMI (13.59%, of which 75.0% concordant) and folding index (15.84%, of which 58.70% concordant). A summary of all the bivariate MiXeR analysis results can be found in Supplementary Table S14.”

We also discussed the sign discordance result as follows (page 19):

“Interestingly, MiXeR analyses showed that a large proportion of variants explaining the heritability in AOW were shared with educational attainment and cognitive performance, with more than half of these variants having concordant effects on the two phenotypes (which explains the overall positive genetic correlations obtained with the LDSC method shown in Fig. 4A). Thus, results indicated that genetic predispositions for later onset of walking also contribute to high cognitive performance and more educational attainment. It is interesting to note that nearly half

of the overlapping SNPs between AOW and cognitive performance and academic achievement have discordant effects.”

Lines 535-536: what does it mean if higher volumes are associated with higher PGS; what is the conceptual interpretation of this statistical results? Same for lines 543-545.

We added some further discussion on the meaning of these results, as requested, on page 19:

“Research on the timing of milestones in prenatal brain development across humans, primates and other mammals shows that longer duration (more prolonged development) is associated with larger brain volumes, and in particular, enlargement of later developing brain structures⁷¹. In line with this, within humans we found that the polygenic score predisposing to later onset of walking is associated with larger volumes of neonatal brain areas involved in the motor domain (Fig. 5). Additionally, we found that gene sets associated with AOW are involved in neurogenesis, and that expression of genes associated with AOW are enriched in the brain between 19 and 24 weeks post conception (Fig. S14). Last, we found that later AOW is genetically correlated with increased cortical folding in adolescence and adulthood in areas involved in the somatosensory processing of movement (ROIs 1 and 5m in Glasser parcellation⁶⁰, located in the primary somatosensory cortex), including higher order somatosensory integration of the lower limb representation (ROI 24dd in the cingulate motor area), and motor planning concerning the whole body (ROI 6r in the premotor cortex). Taken together, these findings may suggest that for children with genetically-influenced protracted subcortical neurogenesis in the prenatal period, cortical regions involved in more complex motor behaviours may take longer to specialise⁷². This results in a later onset of walking. Since advantages and costs to early walking might vary based on the individual’s environmental conditions, wide individual differences in the duration of the sensitive period to learn to walk might be the result of the ability of human beings to adapt to their local environment⁷³.”

The genetic overlap analyses in the paper focus on global genetic correlations. Local genetic correlation analyses, however, can disclose correlated regions for traits that show no significant global genetic correlation. I feel that such local genetic correlation analyses would be an addition to the paper (eg the loci where KANSL1 and RBL2 reside, might be associated to other traits, and diving further into these loci through conditional correlation analyses might further illuminate the specificity of these relations).

Thank you for this suggestion. We have looked into detail into the LAVA method and the implications for adding LAVA analyses here. We have also looked at when and how the

field is using LAVA. It appears to be most common to employ LAVA in separate manuscripts to the original GWAS (e.g., doi.org/10.1038/s41380-024-02510-y; doi.org/10.1038/s44161-024-00488-y; doi.org/10.1038/s41467-023-43567-7). Our concern is that we did not pre-register LAVA analyses and there appear to be a large number of tests we could run. The reviewer also refers to conditional correlation analyses. In an earlier point, they raised concerns about one additional non-pre-registered analysis using GSEM in which we investigated AOW and ADHD after controlling for educational attainment. LAVA would involve many more such non-pre-registered bivariate and multivariate analyses. We are keen not to enter into a situation in which we are picking particular results and not others. In light of the above, it is our suggestion that LAVA analyses are left for a separate manuscript in which the work could be pre-registered and the analysed carefully planned. We note that only one of the four reviewers suggested the addition of LAVA. However, we are happy to take Editorial guidance if it is felt strongly that LAVA analyses should be added into this manuscript.

In response to the reviewer's helpful comment, we have added the following text into the Discussion (page 20): **"Future work could also test the degree to which genetic correlations with AOW vary locally across the genome, and furthermore, how they vary when conditioned on third variables, in order to delineate genetic associations with AOW within specific genomic locations⁷⁶."**

Lines 637-640: this paragraph is understandable mainly for the in-crowd.

Based on this feedback, we have reworded this paragraph to make it more accessible as follows (page 20):

"Our final conclusion is that the genetic signal identified through our AOW GWAS captures genetic effects that directly influence the phenotype⁵¹. This was tested by the within-family polygenic score analyses on fraternal twin siblings in the NTR cohort. We found that variance explained by the between-pair PGS was not significantly greater than variance explained by within-pair association. If the variance explained by between-pair PGS had been much larger than the within-pair PGS, it would have indicated that some of the AOW signal was coming from genetic effects that play a role on the phenotype in an indirect way, via mechanisms such as gene-environment correlation, assortative mating and stochastic effects⁵¹. Our results offer evidence that the polygenic score is picking up on direct genetic effects."

Lines 656-658: sentence doesn't seem to run properly; consider rephrasing.

We have reworded this for clarity (page 20):

“The genetic variants identified were plausible contributors to individual variability in motor behaviour, as they were previously associated with disorders that disrupt the development of walking and are linked to motor disorders.”

Lines 660-662: “Given its high heritability, future research should investigate whether genetic predisposition for age at onset of walking can be incorporated into personalised medicine to inform early intervention strategies for young children.” This links back to the introduction, where the authors also claim that this genetic study could shed light on whether genetic information could be used clinically to distinguish problematic late-walkers from unproblematic ones. I personally feel that given the paper’s results, there is too much emphasis in the introduction on such a clinical application: the authors did not show that the genetic information that we have today is applicable in that sense (unless it is the carrying of certain alleles in 1 or 2 specific genes): if they want to retain this line of reasoning, I find that they need to more clearly work towards that promise. In that same vein, the last sentence of the abstract “This offers new biological insights of clinical relevance” also requires clear reflection in the discussion: what results are exactly clinically relevant and if so, how exactly?

We have reduced the emphasis on clinical application, removed the sentence on personalised medicine, as suggested, and focused more on the relevance of these findings to understand human behaviour.

We also modified the abstract as follows:

“This offers new biological insights into a key behavioural marker of neurodevelopment.”

Supplement:

The genetic correlation between NTR and Lifelines, both Dutch datasets, is quite low and not significant (sample size issue I presume): any thoughts on why the correlation is so low, while the correlation between MoBa and NTR is quite sizable (.89)?

Estimates of the genetic correlation between these two smaller samples (N=3,415 and N=2,592) are not reliable due to low power based on the sample sizes (as pointed out in the LDSC FAQ on “What sample size do I need for LD Score regression?”, indicating “LD Score regression tends to yield very noisy results when applied to datasets with fewer than ~5k” <https://github.com/bulik/ldsc/wiki/FAQ>). Correlation results between the two

larger cohorts (N=58,302 and N=6,251) are more reliable. We made this clear in the SM, on page 13:

“Lower h^2_{SNP} estimates and larger standard errors were obtained for the smaller samples, namely Lifelines ($h^2_{\text{SNP}} = 9.52\%$, SE = 12.98%) and NSHD ($h^2_{\text{SNP}} = -3.02\%$, SE = 17.17%), as LDSC cannot produce reliable estimates with samples < 5K (<https://github.com/bulik/ldsc/wiki/FAQ>). Genetic correlation (r_g) between MoBa and NTR was $r_g = 0.89$ (SE = 0.17, $p = 1.803 \times 10^{-7}$) and between NTR and Lifelines was $r_g = 0.46$ (SE = 0.55, $p = 0.404$). As expected, other genetic correlations were out of bound (MoBa-Lifelines $r_g = 1.168$, SE = 0.72, $p = 0.103$) or non-estimable (MoBa-NHSD r_g , Lifelines-NHSD r_g and NTR-NHSD $r_g = \text{nan}$) due to low reliability of the LDSC estimates, as indicated by the large SNP-based heritability standard errors obtained for the smaller cohorts.”

Many thanks to Reviewer 2.

Reviewer #3:

Remarks to the Author:

This is an exceptionally thorough manuscript by Gui et al. investigating the genetic correlates of age at onset of walking. The authors have meta-analyzed data from over 70,000 children of European ancestry to perform the first genome-wide association study (GWAS) of this phenotype and conducted rigorous follow-up of GWAS findings including biological annotation, colocalization, genetic correlation, and polygenic score (PGS) analyses. Gui et al. establish that age at onset of walking shows significant SNP-based heritability and identify 11 independent genome-wide significant loci. Genes associated with age at onset of walking were enriched for cerebellum eQTLs for RBL2 and KANSL1. Significant genetic correlations were detected with BMI, educational attainment, IQ, ADHD, and neuroimaging measures of cortical folding, and PGS for age at onset of walking predicted infant brain volume.

These novel findings are important for our understanding of child development and may ultimately be useful for identifying children at high risk for developmental delay and facilitating early interventions. The methods employed are appropriate and the authors have done an exemplary job of statistically validating their results. The results are clearly described both in text and in the figures. I have just a few questions and suggestions for where the authors may want to elaborate.

- There seem to be many differences in the way the different cohorts were processed/analyzed (e.g., reference panels, GWAS covariates). What is the justification for these differences and how might it impact results?

Thank you for the opportunity to explain how we approached the GWAS meta-analysis with the aim of reducing differences between the cohorts while including all the cohorts that met our pre-defined quality criteria to maximise the sample size. Of note, to minimise differences in the SNP effects that contributed to the meta-analysis, all cohorts followed a detailed SOP (<https://osf.io/jyk6d/>). The pre-processing choices and analyses were pre-registered before conducting the meta-analysis (<https://doi.org/10.17605/OSF.IO/M2QV3>).

We addressed the key points below.

Genotype data pre-processing

Differences between cohorts in the data processing concerned mainly the pre-imputation genotyping quality control and imputation procedures. These differences are necessary to allow for comparable analytic datasets, since the genotyped cohort data differ in structural ways. For example, some cohorts had relatedness within the sample, duplicate concordance checks and a multi-batch structure of the genotyping which were not relevant to other cohorts (see for example Corfield et al., 2022 <https://doi.org/10.1101/2022.06.23.496289>).

In all cohorts, pre-imputation processing was performed by expert teams of analysts prior to this study. All cohort teams were requested to provide a readme document with key information on their data and pre-processing choices that allowed us to judge if the necessary quality checks were performed prior to imputation. Our post-imputation pre-processing steps, in which the same criteria for MAF, HWE, SNP call rate, imputation quality score filters were applied to all summary statistics, will have removed any minor differences between cohorts.

Imputation reference panel

Imputation was performed using a different imputation panel in NTR (1KG) vs the other three cohorts (HRC). These differences were due to data sharing protocols specific to the individual cohorts but we ensured all panels were on GRCh Build 37 to avoid issues with SNP alignment. Additionally, before running the meta-analysis, all summary statistics were quality checked using GWASinspector (Ani et al., 2021 <https://pubmed.ncbi.nlm.nih.gov/33416854/>), which performed the same additional SNP filtering and alignment procedures on all the cohorts' sumstats to harmonise the data, as detailed in the Methods.

GWAS covariates

In our SOP (<https://osf.io/jyk6d/>), which was used by the research teams of the cohorts contributing to the study, we specified what covariates needed to be included in the individual cohorts' GWAS models, namely "10 PCs, a sex categorical variable with males as the baseline, age at phenotypic data collection (standardised so that the sample mean = 0 and standard deviation = 1), technical control variables

(e.g., genotyping batch or array; if applicable), cohort specific design factors (such as, testing wave, site, rater)”.

The requested covariates were available in most of the cohorts. Sex and the first 10 ancestry principal components were included as covariates for all cohorts. NSHD did not include age at data collection because all the phenotype data were collected at 2 years and no genotyping batch/array variable was available for this cohort. MoBa included the year when the data was collected rather than age at data collection because the data were gathered from all participants at the same age, but the collection year differed, as the cohort was recruited over a span of 10 years. Indeed, our heterogeneity analyses confirmed that there was no heterogeneity between cohorts in the SNP effects.

We now added the reference to the SOP in OSF in the main manuscript (page 22).

“Pre- and post-imputation quality control (QC) and imputation procedures were conducted for each cohort following individual study protocols, and according to a **common** Standard Operating Procedure (<https://osf.io/jyk6d/>), which was based on the Rapid Imputation for COnsortias PipeLine (RICOPILI) pipeline⁸². In all the individual cohorts, samples were excluded from the GWAS if they presented excess autosomal heterozygosity, mismatch between self-reported and genetic sex, XXY genotype and other aneuploidies, individual genotyping rate < 90%. Duplicate samples and samples whose genetically determined ancestry did not overlay with the European ancestry cluster based on a reference panel were also excluded. Autosomal SNPs were excluded from the GWAS if they had minor allele frequency (MAF) < 0.5%, Hardy-Weinberg Equilibrium (HWE) exact test at $p < 1 \times 10^{-6}$, call-rate < 98%. Full details of the pre- and post-imputation QC are provided in the Supplementary Note A and in Supplementary Table S2.”

- The MoBa distribution is shown in Supplementary Fig. S1. Were the distributions for the other cohorts also approximately normally distributed? It would be helpful to see histograms for the other cohorts as well.

Thank you for the suggestion, we have now added the phenotype histograms for all the samples in the Supplementary Notes file (Supplementary Fig. S3, S5 and S7).

- The data for the different cohorts were from parents at different times in children’s development (18 months, 36 months, 2 years, later). What role does recall bias (i.e., time between when child started walking and when data were collected from parents) play in parental report of age of walking across the cohorts? Are there significant differences between cohorts in the reported age at onset of walking and do these map on to the time interval between when children started walking and completion of the parent report?

Parental recall of AOW shows high reliability and validity in early childhood. First, retrospective parental recall within the few years after children begin to walk (which is when all our data were collected), is valid when compared to prospective parental interviews (Langendonk et al., 2007 <https://pubmed.ncbi.nlm.nih.gov/18179395/>). Second, test-retest reliability for parental recall of AOW is very high ($r^2 = 0.84$) across an 18 month window (parent reported age at onset of walking at 18 months and 36 months for N = 50940 children <https://www.ncbi.nlm.nih.gov/pmc/articles/PMC10106302/> see plot below). Finally, a twin study of AOW reported a twin heritability of 84% (Smith et al., 2017 <https://pubmed.ncbi.nlm.nih.gov/29048262/>). This suggests low measurement error, since it is not possible to find high twin heritability in the presence of large measurement error. Measurement error in the twin design is captured as part of non-shared environment term and as such would be under 16%.

NSHD retrospective recalls were collected when children were aged 2 years, similar to the NTR cohort. We acknowledge we do not have the possibility to verify whether the retrospective report is reliable in the Lifelines cohort. However, we do not think recall introduced a bias for this cohort, since the mean reported age and SD ($M=14.27$ ($SD=2.54$)) are in line with those for the other cohorts (MoBa = 12.83 (2.01), NSHD = 13.51 (2.37), NTR = 15.06 (2.36)). We added this point as a possible limitation of this study on page 20:

“Samples were only included if they had a highly similar phenotype (AOW in months) and a sample size of greater than 1000 to ensure reliable effect sizes in individual samples. Nevertheless, the potential attrition and participation biases present in population cohorts should be considered in relation to our findings^{74,75}. Although there is evidence that AOW can be reliably recalled by parents retrospectively by the child’s second¹ and third¹⁹ birthday, we acknowledge that it was not possible to measure the reliability of this phenotype as recorded in the Lifelines cohort, where it was collected between the children’s 3 and 18 years of age. It is possible that the Lifelines measure included the most measurement error of the four cohorts, in light of the later age at which parents recalled the AOW in their children.”

We did not find a significant or large correlation between phenotypic mean differences between cohorts and the differences in the interval between AOW and parent report ($r=0.16$, $p=0.76$).

We have not added text on the exact causes of the mean differences in AOW between cohorts in the manuscript. There are a range of factors that are likely involved. Year of birth may play a role: there is a time trend for infants to walk slightly later in more recent years than past years. Systematic regional/national differences in body size and cultural factors might add to the mean difference in AOW between cohorts. By conducting the GWAS in each individual cohort and then meta-analysing the effects, our approach ensures to capture SNP effects on the trait variance that are not confounded by differences between cohorts. Since mean differences in AOW are not the focus of this manuscript, we have not added a detailed discussion on this point but are happy to take Editorial guidance on this if it is considered important to add in.

- The authors report I^2 as an index of genome wide heterogeneity between cohorts. I^2 provides a measure of heterogeneity on an individual SNP basis. Were there any patterns of multi-SNP heterogeneity (M)?

We now added the multi-SNP heterogeneity statistics results in the main text (page 8) and Supplementary Table S3:

“Overall, the M multi-SNP heterogeneity metric across the independent lead SNPs³⁴ associated with AOW indicated no systematically more or less influential study (see Supplementary Table S3, all Bonferroni-corrected p s < 0.401).”

We also added information about the heterogeneity metrics calculations in the Methods (page 23):

“The I^2 heterogeneity metric per SNP was calculated in METAL. M multi-SNP heterogeneity statistics indicating whether individual studies were systematically more influential or weaker than average based on their effects was calculated using the getmstatistic R package for the independent lead SNPs (pairwise LD $r^2 < 0.1$, $p < 5 \times 10^{-8}$, N SNPs = 15)³⁴.”

- I understand that the cerebellum showed the highest enrichment for expression of genes associated with age at onset of walking, providing justification for focusing on colocalization with RNA seq data from the human cerebellum (N=261). However, there also seems to be enrichment for gene expressed in the cortex (Supplementary Fig. S10), and there may be better powered publicly available RNA-seq data from the adult cortex that could be used for colocalization analyses as well.

As requested, we performed additional colocalization analyses using cortical eQTL data from Sieberts *et al.*, 2020. For *RBL2*, this analysis identified the same missense variant (p.Tyr210Cys) as the likely causal variant in the GWAS, with two distinct groups as in the analyses based on the cerebellum. We have added this result to the Results (page 10):

“A similar colocalization pattern was observed using 1,433 samples of the human adult cortex⁴³ (Supplementary Fig. S16; posterior probability = 0.97 - 0.99).”

With greater statistical power from the cortex eQTL data, we observed that SNPs associated with age of onset of walking were also associated with expression of several genes in genomic locus 6 on chromosome 17. This may be due to the complex haplotype structure and alternative contigs in this region. As a result, we have de-emphasised the colocalization result in *KANSL1* from the cerebellum eQTL data, as several genes show colocalization in this region, making it difficult to identify the causal gene. We have modified the text in the Results (page 12) to reflect these changes:

“We also identified colocalization of SNPs associated with expression of several genes in both the cerebellum and cortex with SNPs associated with AOW in genomic locus 6 on chromosome 17 (Table 1). This region has a complex haplotype structure, including alternative contigs, which may explain this result. In cerebellum, we identified colocalization in *KANSL1* (PP=0.79), *PLEKHM1* (PP=0.78), *SPPL2C* (PP=0.77) and *STH* (PP=0.63). In cortex, we also identified colocalization in *STH* (PP=0.78) and *SPPL2C* (PP=0.72), as well as *CRHR1* (PP=0.74).”

We also removed the paragraph on *KANSL1* in the Discussion to reduce emphasis on these results. We thank the reviewer for suggesting to perform this additional analysis.

The Figure below illustrating *RBL2* colocalization results in the cortex has been added to the Supplementary Notes as Supplementary Fig. S16.

- Colocalization was performed with RNA-seq from the cerebellum. Is the RBL2 expression shown in Figure 2D from cerebellum or cortex? I believe BrainVar data is from the DLPFC. Is there cerebellar RNA-seq data that could be used to show expression by genotype?

Apologies for the confusion, Fig. 2D shows colocalisation results using RNA expression from the prefrontal cortex, we now clarified this in the figure caption.

“D. Swarm, violin, and box-plots showing the distribution of *RBL2* expression in prefrontal cortex (transcripts per million (TPM), y-axis).”

We are not aware of any datasets that have cerebellar RNA-seq data and individual level genotype on the same individuals, therefore to validate our colocalisation results on the cerebellum we used RNA expression in the cortex using a sample that had available RNA-seq and genotype data (prefrontal cortex from BrainVar). We now clarified this in the Methods, page 24:

“The eQTL data used in the colocalization analyses were from 261 post-mortem bulk RNA-seq samples of human cerebellum⁴³. We replicated the colocalization signal observed in *RBL2* (Fig. 2B) in the human cortex using eQTL data from 1,433 post-mortem bulk RNA-seq samples⁴³ (Supplementary Fig. S16). To validate, in an independent data set whether genotype was indeed associated with *RBL2* expression, we used bulk-RNA-seq data of prefrontal cortex and individual-level genotypes from

BrainVar⁴⁵ (periods 4-6; Fig. 2D) (as no publicly available cerebellum RNA-seq with genotype on the same individual exists, to our knowledge).”

- It appears that all genetic correlations with neuroimaging measures were for the whole brain. Have the authors considered, for whole-brain genetic correlations that survived Bonferroni correction (i.e., folding index), testing genetic correlations with local cortical folding for brain regions of interest (ROIs) to gain more specificity as to the specific brain regions implicated?

Thank you for this suggestion, we now added the results to this analysis in the Supplementary Table S15 and mentioned it in the main manuscript.

Results (page 13-14):

“In light of our findings of a Bonferroni-significant genetic correlation between AOW and global folding index, we conducted further non pre-registered analyses, as requested by a reviewer, to gain more specific information about the brain regions implicated. We included regions involved in motor and/or somatosensory function and corrected for multiple testing using FDR correction. We found that later AOW was significantly genetically correlated with increased folding in the primary somatosensory cortex (ROIs 1 and 5m in Glasser parcellation⁶⁰) premotor cortex (ROI 6r) and cingulate motor area (ROI 24dd, see Supplementary Table S15 for the full set of results).”

Discussion (page 19):

“Research on the timing of milestones in prenatal brain development across humans, primates and other mammals shows that longer duration (more prolonged development) is associated with larger brain volumes, and in particular, enlargement of later developing brain structures⁷¹. In line with this, within humans we found that the polygenic score predisposing to later onset of walking is associated with larger volumes of neonatal brain areas involved in the motor domain (Fig. 5). Additionally, we found that gene sets associated with AOW are involved in neurogenesis, and that expression of genes associated with AOW are enriched in the brain between 19 and 24 weeks post conception (Fig. S14). Last, we found that later AOW is genetically correlated with increased cortical folding in adolescence and adulthood in areas involved in the somatosensory processing of movement (ROIs 1 and 5m in Glasser parcellation⁶⁰, located in the primary somatosensory cortex), including higher order somatosensory integration of the lower limb representation (ROI 24dd in the cingulate motor area), and motor planning concerning the whole body (ROI 6r in the premotor cortex). Taken together, these findings may suggest that for children with genetically-influenced protracted subcortical neurogenesis in the prenatal period, cortical regions involved in more complex motor behaviours may take longer to specialise⁷². This results in a later onset of walking. Since advantages and costs to early walking might vary based on the individual’s environmental conditions, wide individual

differences in the duration of the sensitive period to learn to walk might be the result of the ability of human beings to adapt to their local environment⁷³.”

Methods (page 25):

“In order to further investigate the significant genetic correlation between AOW and cortical folding index, we ran non pre-registered genetic correlation analyses using 26 regional FI summary statistics from ⁵⁸. The 26 regions of interests (ROIs) were based on the Glasser parcellation and identified based on their functional specialisation as early somato-sensory/motor areas according to ⁶⁰. Given that regional FI could not be assumed to be completely unrelated, we applied false discovery rate (FDR) correction of 26 tests.”

- As the most robust evidence for genetic correlation was found between age at onset of walking and folding index, it is not clear why the authors tested associations with brain volume in the Developing Human Connectome Project rather than folding/gyrification.

Thank you for the suggestion to include folding phenotypes in the Developing Human Connectome Project analyses. We have now added this analysis to the manuscript and updated to Results section on page 16:

“To explore whether the correlation between gyrification and common genetic variation linked to AOW was present in newborns, we fit a GLM testing for a significant effect of AOW PGS on the mean gyrification index in the left and right hemisphere of the dHCP infants. We found a significant positive association between AOW PGS and gyrification index in both hemispheres in newborn brains ($q < 0.05$).”

And the Methods section on page 27-28:

“Imaging data: Acquisition, processing and surface generation

T2-weighted MRI data were acquired at term equivalent age (median postmenstrual age = 41.9 weeks) as part of the Developing Human Connectome Project (dHCP⁶³). The volumes were run through the neonatal-specific processing pipeline developed for the dHCP study, including bias field correction, brain extraction and image segmentation¹¹⁶⁻¹¹⁸. Segmentations were used to generate cortical, white matter and pial surfaces, and each subject was visually inspected to ensure accuracy, before the local gyrification index was calculated at each vertex, based on the ratio of the pial and white matter surface area^{119,120}.”

Reviewer #4:

Remarks to the Author:

NMED-A133391

Age of onset of walking (AOW) is a developmental milestone linked to normative and abnormal development and delays are an indicator of potential developmental delay. In this manuscript, the authors report the first GWAS meta-analysis of age at onset of walking in a large ($N > 70K$) sample of children from 4 European ancestry cohorts (from northern Europe and the UK). In each cohort, the phenotype was based on a single question to parents regarding the age at which their child first walked without assistance. The authors find that age at onset of walking (AOW) is significantly heritable ($h^2_{SNP} = 24\%$) and polygenic. They identify 11 independent genome-wide significant loci most in or near genes with plausible relevance to the phenotype. Using fine mapping and gene expression data, they show that one of these appears to be a haplotype associated with decreased expression of *RBL2* and encompassing a potentially deleterious missense variant in that gene. Colocalization analysis also implicates a causal variant at and reduced expression of *KANSL1*. Of note, prior work has shown that mutations in these genes are also causal for severe neurodevelopmental disorders that include delayed AOW. Follow-up functional genomic analyses showed enrichment of signals exclusively in brain tissues, especially cerebellum, basal ganglia, and cortex. PGS analyses suggest that the common genetic effects are mostly direct (rather than due to gene-environment correlation or assortative mating) and link genomic variation to volumes of specific brain regions. Genetic correlation analyses demonstrate indicate positive correlations with educational attainment and IQ, and negative correlation with ADHD and BMI—with other, lesser positive correlations with bipolar disorder, cortical folding and gyrification.

The manuscripts has several notable strengths. Analyses are comprehensive and rigorous, and the bioinformatic analyses provide valuable insights into potentially causal variation and tissue expression of implicated genes. The use of roughly contemporaneous reports to assess AOW is a strength. The meta-analysis is dominated by the MoBa sample which showed substantial inflation of lambda, but the authors performed extensive secondary analyses to support the conclusion that this reflects polygenicity rather than spurious inflation e.g. due to residual population stratification, relatedness, or outliers.

The analyses were restricted to European ancestry samples, but the authors appropriately highlight this weakness. In terms of other weaknesses and areas to address:

1. My main comment is that the relevance of the work to the journal's scope of

“impact on improving human health by addressing unmet clinical needs”. The authors do note that AOW is relevant to pediatric evaluation of neurodevelopmental milestones as indicators of potential abnormalities and the genetic correlation analyses suggest etiologic overlap with clinical syndromes (ADHD, ASD, bipolar disorder). However, the implications seem under-developed and include generic statements such as that the results “can contribute alongside screening tools to aid the prediction and early identification of clinically-relevant conditions associated with early or delayed onset of walking, and avoid missing time for potentially beneficial physical training when appropriate.” And “A better understanding of the entire variation of age at onset of walking and of its shared biology with later medically-relevant phenotypes can help in more precise intervention planning.” Given that the findings point to a highly polygenic trait for which PGS explain a small proportion of variation, it is unclear how they might actually be clinically useful beyond current assessments. It may be beyond the scope of this paper, but one could evaluate whether in fact adding PGS to modeled clinical variables (and/or rare variation) that might be available in the data provide meaningful improvements in predicting diagnosis or development.

The manuscript is now being considered by Nature Human Behavior. In light of the change of journal and the reviewer’s comments, we reduced the emphasis on clinical application. Tests for clinical application are an important next step which will mention in the Discussion.

Abstract:

“This offers new biological insights into a key behavioural marker of neurodevelopment.”

Introduction:

We REMOVED the sentence: “In addition to the resulting healthcare costs for assessment and investigation, many parents will also experience undue stress if their child’s late walking leads to clinical follow up in the context of normal variation, i.e. the child’s late walking is not signalling clinical need per se, but just their predisposition for later walking.”

Discussion (page 20):

“A better understanding of the entire variation of AOW and of its shared biology with later medically-relevant phenotypes could help in more precise intervention planning. Future research should test whether adding AOW PGS to clinical variable and/or rare variant information could improve prediction models that could be applied clinically.”

We have not added in clinical modelling for two reasons. First, it is not possible to use PGS in the samples from which the GWAS was derived (target samples need to be independent). Second, the manuscript is no longer being considered by Nature Medicine.

2. The inverse genetic correlation between AOW and ADHD (and lack thereof for ASD) is intriguing as is the positive r_g with IQ and educational attainment. Delayed AOW has been associated with intellectual disability and to a lesser extent ASD. To the extent that ADHD is an NDD, the negative r_g seems surprising. The authors might comment on this, particularly given the claimed relevance of their analyses for insights into neurodevelopment.

We now added further discussion on this point of interest on page 19:

“The negative genetic correlation between AOW and ADHD might be surprising when considering that delayed walking, rather than earlier walking, is associated with increased likelihood of developmental disorders⁵. However, the ability to walk requires practice and movement⁶⁶, and infants with higher activity levels or shorter attention spans may — on average — move about more, thus gaining more practice in movement and muscle strengthening and training, ultimately resulting in earlier walking onset. Thus, attention and activity levels may influence motor system training in young children. In support of the hypothesis that shorter attention span and higher activity levels would be associated with earlier walking, a recent study of over 25,000 children from MoBa found that the ADHD polygenic score was associated with earlier walking¹⁹. Further, the ADHD polygenic score was associated with better gross motor skills, such as walking, climbing stairs, and jumping, in 7,498 18-month- old children from the Avon Longitudinal Study of Parents and Children (ALSPAC)⁷⁰. At the same time, it should be noted that in our study the negative genetic correlations between AOW and ADHD is, while significant, still relatively modest in magnitude ($r_g = -0.19$).”

3. If I am understanding correctly, the eQTL analyses relied on (bulk) RNA-seq data from cerebellum, and only significant eQTLs from this analysis were used in colocalization analyses and carried forward for analyses of cortex. Could the authors comment on the justification for this choice?

We thank the reviewer for this question and the opportunity to clarify our decision-making process.

As the reviewer points out, eQTL analyses were conducted in cerebellum by Sieberts *et al.*, 2020 <https://doi.org/10.1038/s41597-020-00642-8> . The authors of this study provided publicly available summary statistics for the association of SNPs with gene expression in cerebellum, including betas, p values and standard errors. These

summary statistics, along with the summary statistics from our own GWAS on age at onset of walking, were used to perform colocalization analysis using coloc SuSiE. Coloc SuSiE considers all p values and standard errors (along with sample size and LD) in a region to perform colocalization analysis, and does not necessarily only consider significant eQTLs (though significance of the SNPs is a factor). Of note, we get very similar results using Sieberts cortex eQTLs, too - see also our response to Reviewer 3 above.

We sought to validate, in an independent data set with RNA-seq data, whether one's genotype for SNPs in Group 2 (dark blue), were indeed associated with lower *RBL2* expression (as seen in Sieberts *et al.*). Unfortunately, to our knowledge, publicly available datasets to test this, with both bulk-RNA from cerebellum and individual-level genotypes, do not exist. However, we did have access to data with bulk-RNA from cortex and individual-level genotypes, which we used as validation instead.

We are aware this is not the same as replicating in an independent dataset with RNA-seq from cerebellum, however:

- 1) We have now demonstrated the same colocalization signal in cortex, included as Supplementary Fig. S16.
- 2) In addition to cerebellum, we also identified enrichment for expression of genes associated with age at onset of walking in cortex (Supplementary Fig. S15).
- 3) Generally, eQTLs across brain regions are highly correlated and overlapping (The GTEx Consortium, 2015 [10.1126/science.1262110](https://doi.org/10.1126/science.1262110), 2017 [10.1038/nature24277](https://doi.org/10.1038/nature24277), 2020 [10.1126/science.aaz1776](https://doi.org/10.1126/science.aaz1776)).

We now clarified this choice in the Methods, on page 24:

“The eQTL data used in the colocalization analyses were from 261 post-mortem bulk RNA-seq samples of human cerebellum⁴³. We replicated the colocalization signal observed in *RBL2* (Fig. 2B) in the human cortex using eQTL data from 1,433 post-mortem bulk RNA-seq samples⁴³ (Supplementary Fig. S16). To validate, in an independent data set whether genotype was indeed associated with *RBL2* expression, we used bulk-RNA-seq data of prefrontal cortex and individual-level genotypes from BrainVar⁴⁵ (periods 4-6; Fig. 2D) (as no publicly available cerebellum RNA-seq with genotype on the same individual exists, to our knowledge).”

Many thanks to Reviewer 4.

General:

This study concerns a genome-wide meta-analysis of 4 cohorts of age-at-onset-of-walking. The analyses are quite comprehensive and the study is interesting. I do have a few comments.

Lines 164-173: This summary of results seems a bit out of place: I would expect it in the abstract or the discussion but not here as such?

We have provided a short summary of the key findings at the end of the introduction, as done in similar papers, to give the reader a clear idea of the whole research before diving into the Results section. Our preference would be to leave the current text, but we have prepared an alternative paragraph and we will take editorial guidance on whether the current text should be replaced with the following:

“Here we present a genome-wide association study (GWAS) meta-analysis of AOW in a sample of 70,560 children from four European-ancestry cohorts. First, we aimed to quantify SNP-based heritability of AOW and the degree of polygenicity of this trait. Second, we aimed to identify independent genetic loci, and their functional roles. Third, we estimated genetic correlations with physical health indicators, cognitive traits, neurodevelopmental conditions, psychiatric disorders and cortical phenotypes. Fourth, we evaluated the predictive power of the AOW polygenic score and tested whether it was associated with the volume of neonatal brain structures in an independent cohort.”

Line 197 and on: I believe it is customary to not only mention the number of independent loci but also the overall number of genome-wide significant SNPs?

Thank you for the suggestion, we have now added the total number of significant SNPs in the text: “We identified 2,525 genome-wide significant SNPs ($p < 5 \times 10^{-8}$), of which 11 were independent loci with one lead variant per locus in GCTA conditional and joint analysis (COJO)³⁰ (Table 1, Fig. 1, see also Supplementary Fig. S9 for the QQ-plot and Fig. S10 for the regional plots).”.

Lines 232-236: I like the male/female distinction. Is it really the case that h^2_{SNP} was 23.06 for both yet 24.13 combined?

Yes, we double checked the results and these are indeed correct. The larger SNP heritability estimate in the whole sample likely emerges from doubling the sample

size of the GWAS, which allows for more power in detecting the contribution of genetic variants to the trait.

Line 242-243: I would mention the h^2_{SNP} per sample and the r_{GS} between samples (Suppl note C) at the start of this paragraph as it is informative on whether we should meta-analyse these cohorts at all?

We have moved supplementary note C to the main text as suggested. While the LDSC software cannot provide reliable heritability and genetic correlation estimates for the cohorts with $N < 5K$, this does not indicate individual SNP effects cannot be estimated reliably. In fact, our heterogeneity and quality control analyses demonstrate the smaller cohorts did not provide unreliable information at the level of SNP effects and could therefore be included in the meta-analysis.

Fig 1: legend says inflation factor = 1.27, while the text says 1.23 (line 187) and LDSC intercept of 1.00 while the text says 1.008 (line 188; rounded would be 1.01).

While line 187 refers to the λ_{GC} and intercept for the MoBa cohort, which is the only cohort where the λ value indicates inflation, the figure reports the inflation factor and intercept for the GWAS meta-analysis.

Line 238: sentence doesn't run properly.

We rephrased the sentence as follows: "There was no genome-wide statistically significant heterogeneity (using the conventional $p < 5 \times 10^{-8}$ threshold) between cohorts as tested with I^2 ; the maximum I^2 was 95.3 for SNPs rs7864115 ($\chi^2(1) = 21.453$, $p = 3.63 \times 10^{-6}$) and rs148684045 ($\chi^2(1) = 21.441$, $p = 3.65 \times 10^{-6}$), respectively."

Line 269: inconsistent use of . and , in larger numbers (please check throughout: I also saw instances in the supplemental tables).

Thank you for the notice, we carefully checked this throughout the manuscript.

Line 277: sentence doesn't run properly ("we found THAT...")

We have corrected the sentence as suggested: "Furthermore, we found that 7 of the 47 genes were associated with autism (14.9%), which is over four times the proportion of autism-associated genes in the panel (v0.36, largely based on SFARI-gene⁴⁰) as a whole (734 out of 19,950, 3.68%; $\chi^2(1) = 13.7$; $p = 0.0002$, two-tailed)."

Line 409: the p-value of .70 does not seem to fit the chi-sq value of .04 (in R: `1-pchisq(.04,1)` gives .84).

Thank you for noticing this, we now corrected the statistics as follows:

“Among 1,254 dizygotic twin pairs (N = 2,508 individuals), within- and between-family standardized regression coefficients in a linear mixed-effects model were not significantly different from each other ($\chi^2(1) = 1.185$, $p = 0.276$, two-tailed), indicating that the genetic signal is not biased by prGE, or effects such as stratification and assortative mating”.

Line 411: seems to fit more with the previous paragraph.

We are introducing Fig. 3 at this point to allow the reader to appreciate the graphical representation of the results of the within- and between sib-pair analysis as well. We clarified this in the text as follows:

“Fig. 3 shows the beta estimates of the AOW PGS prediction in all the cohorts, with the NTR within- and between- sib-pair estimates presented separately.”

Line 451: “of AOW” I think, not “with AOW”.

We corrected this line as follows:

“The largest magnitude genetic correlation was between AOW and ADHD.”

Lines 475-486: what about ADHD? It seems to odd to single out ADHD in the previous paragraph (451-459) and not discuss it here? Also: the SNP-overlap with educ attainment and cognition seems very high and the discordance signal quite substantial: would it make sense to run LAVA on these traits and see whether maybe some local correlations are negative, some positive, cancelling each other out in the global LDSC rg ? (609-611 also notes explicitly that these discordant effects are of interest: LAVA allows one to study these discordant effects specifically).

Thank you for the interesting suggestion. In line with editorial guidance, we are not including these analyses in the current manuscript but look forward to exploring these in future research.

Line 566: delete “that” (“2 of which that colocalised”).

We deleted “that” as suggested.

Line 596 “our third conclusion from our results”: unfortunately phrased. I find “our xx conclusion” somewhat less formal anyways.

We rephrased this sentence as follows:

“The third conclusion from our results is that AOW is partly influenced by the same genetic variants that influence individual variability of other complex traits measured at later ages.”

Lines 598 and on: be careful not to interpret or phrase these genetic correlations as if they were phenotypic.

We rephrased this sentence as follows:

“We found that common genetic variation associated with later AOW is partly overlapping with common genetic variation associated with higher cognitive performance and more years in education, lower likelihood of ADHD, higher folding index, local gyrification index, cortical surface area and isotropic volume fraction.”

Line 613 -619: genetic correlations interpreted very much in phenotypic terms, please consider careful rephrasing.

We have now highlighted the distinction between phenotypic and genetic correlation.

“The negative genetic correlation between AOW and ADHD might be surprising when considering that, at the phenotypic level, delayed walking, rather than earlier walking, is associated with increased likelihood of developmental disorders⁵. However, the ability to walk requires practice and movement⁶⁵, and infants with higher activity levels or shorter attention spans may — on average — move about more, thus gaining more practice in movement and muscle strengthening and training, ultimately resulting in earlier walking onset. Thus, attention and activity levels may influence motor system training in young children, and this may relate to what we are observing at the level of common genetic variation.”

Line 627: I believe previously this rg was -.18, not -.19?

Thank you for pointing out this typo, we now corrected the text as follows:

“At the same time, it should be noted that in our study the negative genetic correlations between AOW and ADHD is, while significant, still relatively modest in magnitude ($r_g = -0.180$).”.

Lines 642-643: “that for children with genetically-influenced protracted subcortical neurogenesis in the prenatal period”; are there also children without genetically-influenced subcortical neurogenesis?

General: I see, 1, 2 and 3 decimals being used: I would opt for 2 throughout, not a mix?

We are now reporting three decimals, as indicated in the editorial guidelines.

Suppl Table 1:

there seem to be considerable mean differences in AOW between the 4 samples; can these be explained?

Since a similar point was raised by another reviewer, we considered it preferable to add our response to this comment in the main text to address possible concerns in the readers. In the limitations section, we added the text in yellow:

“Although there is evidence that AOW can be reliably recalled by parents retrospectively by the child’s second¹ and third¹⁹ birthday, we acknowledge that it was not possible to measure the reliability of this phenotype as recorded in the Lifelines cohort, where it was collected between the children’s 3 and 18 years of age. It is possible that the Lifelines measure included the most measurement error of the four cohorts, in light of the later age at which parents recalled the AOW in their children. Of note, the interval between AOW and parent report was not significantly correlated with mean AOW difference between cohorts ($r=0.16$, $p=0.76$, two-tailed). Systematic regional/national differences in body size and cultural factors might explain these differences. However, by conducting the GWAS in each individual cohort and then meta-analysing the effects, our approach ensures to capture SNP effects on the trait variance that are not confounded by differences between cohorts.”

Suppl Table 2:

* inconsistent use of . and , in larger numbers.

Thank you for the note, we checked this throughout the manuscript and SM.

* the sample sizes before QC are much much larger, especially for cohort 1, 2 and 4: I presume participants dropped out because they did not have phenotypic information of AOW and not all because of genetic QC reasons? I would suggest to report the N with valid genotype/phenotype data in that first column, and not all people in the database (if that is what the authors did: not sure really).

Thank you for suggesting this was unclear. We now added a column in the Supplementary Table S2 indicating how many of the original samples for each cohort had phenotypic data.